# Caption This, Reason That: VLMs Caught in the Middle

**Zihan Weng**[*]
Integrated Program in Neuroscience (IPN)
McGill University
Mila, University of Montreal
Canada
`zihan.weng@mail.mcgill.ca`

**Lucas Gomez**[*]
Integrated Program in Neuroscience (IPN)
McGill University
Mila, University of Montreal
Canada
`lucas.gomez@mail.mcgill.ca`
`https://www.lucasgomez.ca/`

**Taylor Whittington Webb**
Microsoft Research
USA
`taylor.w.webb@gmail.com`

**Pouya Bashivan**
Department of Physiology
McGill University
Mila, University of Montreal
Canada
`pouya.bashivan@mcgill.ca`

## Abstract

Vision-Language Models (VLMs) have shown remarkable progress in visual understanding in recent years. Yet, they still lag behind human capabilities in specific visual tasks such as counting or relational reasoning. To understand the underlying limitations, we adopt methodologies from cognitive science, analyzing VLM performance along core cognitive axes: Perception, Attention, and Memory. Using a suite of tasks targeting these abilities, we evaluate state-of-the-art VLMs, including GPT-4o. Our analysis reveals distinct cognitive profiles: while advanced models approach ceiling performance on some tasks (e.g. category identification), a significant gap persists, particularly in tasks requiring spatial understanding or selective attention. Investigating the source of these failures and potential methods for improvement, we employ a vision-text decoupling analysis, finding that models struggling with direct visual reasoning show marked improvement when reasoning over their own generated text captions. These experiments reveal a strong need for improved VLM Chain-of-Thought (CoT) abilities, even in models that consistently exceed human performance. Furthermore, we demonstrate the potential of targeted fine-tuning on composite visual reasoning tasks and show that fine-tuning smaller VLMs moderately improves core cognitive abilities. While this improvement does not translate to large enhancements on challenging, out-of-distribution benchmarks, we show broadly that VLM performance on our datasets strongly correlates with performance on established benchmarks like MMMU-Pro and VQAv2. Our work provides a detailed analysis of VLM cognitive strengths and weaknesses and identifies key bottlenecks in simultaneous perception and reasoning while also providing an effective and simple solution.

---

[*]Equal contribution.

39th Conference on Neural Information Processing Systems (NeurIPS 2025).

# 1  Introduction

A hallmark of human intelligence is the capacity for logical reasoning to solve problems and make decisions. Replicating these abilities in artificial beings has been a longstanding goal in the field of artificial intelligence. Recent advancements have demonstrated that large language models can sometimes exhibit reasoning capacity deemed comparable to humans, functioning effectively as few-shot or even zero-shot reasoners [1–4]. Innovations such as chain-of-thought (CoT) prompting and majority voting have further enhanced these models, enabling them to approach, and in some cases rival, human-level reasoning capabilities, as evidenced by their performance on tasks like coding challenges and the ARC benchmark [5–7].

Building on the success of large language models (LLMs), vision-language models (VLMs) have emerged to address vision-language tasks. These large-scale models integrate pre-trained LLMs with vision models, such as Vision Transformers (ViTs), enabling them to proficiently handle tasks like visual question answering and scene description [8–14].

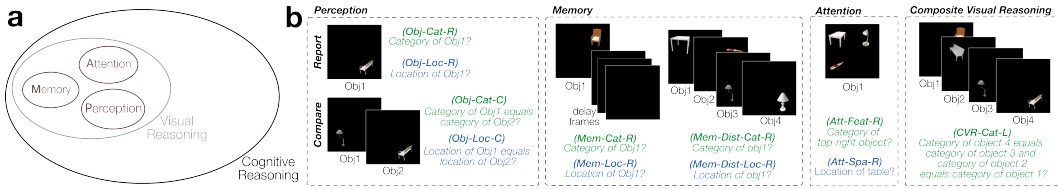

Figure 1: **PAM Dataset.** a) Relationship between different cognitive abilities underlying reasoning. b) Different components of the PAM dataset and example tasks from each. Green and blue correspond to Category and Location tasks respectively. A complete list of examples is provided in Appendix Figures A.1.1-A.1.5.

State-of-the-art (SOTA) vision-language models (VLMs) have been evaluated across a wide range of benchmarks spanning diverse task domains such as simple object recognition, document understanding, and general reasoning tasks [15–18]. Prominent benchmarks like MMMU-Pro, MMBench, and MME primarily assess cognitive reasoning at a high level, focusing on specific tasks such as coding and mathematical problem-solving [19–21].

Despite their strengths, VLMs continue to struggle with visual decision-making and reasoning tasks [22–25]. Campbell et al. [26] show that even state-of-the-art models perform poorly on multi-object reasoning, such as counting or identifying objects, highlighting persistent deficits in visual binding, a key perceptual ability [27]. Spatial reasoning remains another unsolved VLM challenge [28–30, 23, 31].

Evidently, most of these VLM evaluations ultimately deviate from human cognitive science, where researchers break intelligence down into core facultiesPerception, Attention, and Memorythat underlie higher-level functions like reasoning, decision-making, planning, and, more broadly, intelligence [32–35]. In humans, Perception refers to fine-grained sensory encoding [36], Memory to maintaining information despite distraction [37], and Attention to selecting goal-relevant inputs [38]. By reframing VLM evaluation around these core abilities, we can pinpoint which cognitive mechanisms fail and whether deficits in tasks like spatial reasoning arise from encoding, maintenance, or selection. Here, we adopt that perspective and probe various VLMs through the lens of cognitive science.

Our contributions are as follows:

1. Systematically analyzing the cognitive profiles of state-of-the-art VLMs using the procedurally-generated Perception-Attention-Memory (PAM) and Composite Visual Reasoning (CVR) datasets.

2. Identifying specific weaknesses, particularly the pervasive difficulty with spatial information (localization) and selective attention, even in top-performing models.

3. Investigating the nature of the reasoning bottlenecks using a VLM-only vision-text decoupling paradigm, revealing that limitations often stem from the integration of visual information rather than solely from perceptual encoding or language-based reasoning.

4. Demonstrating that targeted fine-tuning on diverse cognitive reasoning tasks can significantly enhance core cognitive abilities in VLMs, while also generally validating that improvements on PAM and CVR tasks translate to broader gains in visual reasoning performance.

## 2 Related Works

The rapid development and success of large transformer-based language models have inspired researchers to extend these architectures and scaling frameworks to other multi-task domains. Many of these domains involve multi-modal data, including visual imagery, video, audio, real or virtual sensory inputs, and language. Integrating large language models (LLMs) directly into VLMs has shown major success across various multi-task vision-language applications [39–42, 14, 12].

Shortly after the release of early VLMs such as GPT-4 Vision, Gemini, Flamingo, and the original LLaVA [10, 8, 43, 42], researchers began assessing their vision-language capabilities. Commonly evaluated skills include image classification, captioning, scene description, and visual question answering (VQA). Datasets like InfoVQA and VQAv2 focus on visual understanding grounded in general world knowledge [15, 44, 45, 18], while others target specialized domains, testing models on scientific reasoning, mathematics, and document comprehension [16, 46, 17, 47].

While some of these datasets involve reasoning about visual information, more complex reasoning and cognitive evaluations have since emerged to measure these features more extensively. Prominent examples include MMMU, MMBench, and MME [19–21]. These benchmarks adopt a comprehensive approach to evaluating reasoning and cognitive skills in state-of-the-art VLMs. MME and MMBench categorize their tasks into reasoning and perception, with both groups sampling diverse contextual domains. However, these benchmarks fail to rigorously evaluate low-level cognitive abilities necessary for visual reasoning.

Beyond these broad vision-language benchmarks, recent studies have focused on testing VLMs on more narrow cognitive domains. BlindTest [48] measures the abilities of VLMs to solve visual geometric tasks. VCog-Bench [49] takes inspiration from early neuro-developmental cognitive tests and evaluates VLMs on their ability to solve abstract logic puzzles and pattern recognition tasks. VisFactor from [50] measures similar highly abstract visual reasoning abilities. Schulze Buschoff et al. [51] focus their cognitive evaluations on intuitive physics, causal reasoning, and intuitive psychology. Visual abductive reasoning is another related area of research[52, 53]. It focuses on the inference of a hidden cause for a give observation, which differs from our focusing on logical reasoning. However, by far the most commonly measured cognitive domain is spatial reasoning [28–30, 23, 31].

While all of these studies highlight important weaknesses within specific domains, they fall short of providing a comprehensive evaluation of VLMs' cognitive visual reasoning. Specifically, they (1) fail to assess core cognitive abilities and their interrelations; (2) focus mainly on single- or few-image tasks; and (3) rely heavily on abstract stimuli that are likely underrepresented in VLM training data. Our work addresses these weaknesses through a systematic evaluation of Perception, Memory, and Attention, their integration in complex visual reasoning tasks, and introduces methods to improve VLM visual reasoning abilities overall.

## 3 Methods

### 3.1 Data Source

We utilize the iWISDM task environment [22] to generate all cognitive tasks and fine-tuning data in this study. This environment enables the procedural generation of an effectively limitless number of vision-language decision-making tasks. In this study, we leverage iWISDM to generate tasks with varying levels of complexity, aligning with the requirements of each cognitive axis in the PAM dataset. These tasks range from simple single-object localization to more complex ones requiring logical reasoning and object comparisons across image sequences. The PAM dataset consists of tasks manually designed to isolate core cognitive abilities. The CVR dataset was generated using

iWISDM to produce complex, composite reasoning problems. Together they provide a wholistic set of tasks that allow us to systematically analyzing the cognitive profiles of state-of-the-art VLMs.

We used ShapeNet objects [54], which include images of 3D-rendered everyday objects taken at various viewing angles. There are 8 object categories and 8 unique objects for each category, and objects are placed in one of four possible locations: top left, top right, bottom left, and bottom right. While not fully naturalistic, these familiar objects enable the measurement of low-level cognitive abilities in a controlled setting, while largely avoiding the issue of limited training-data representation that affects more abstract stimuli such as simple geometric shapes used in prior work [49, 50].

### 3.2 Cognitive tasks for Perception, Attention, & Memory (PAM)

The PAM dataset includes three categories of tasks which measure individual cognitive abilities: Perception (Perc), Attention (Att), and Memory (Mem) (Figure 1).

- **Perception**. The Perception tasks assess a model's immediate access to specific visual object properties according to the task instruction. These tasks contain one or more object frames and require the agent to be able to identify visual object properties, such as spatial location.

- **Attention**. The Attention tasks assess a model's ability to select the task-relevant object from multiple distractors within individual image frames. There are two variants of attention-based tasks: (1) Spatial attention tasks, where the target is specified by its location (e.g. "top-right"); (2) Feature attention tasks where the target is specified by its category (e.g. "chair"). In both cases, the model must ignore irrelevant objects and report or compare the cued object's property.

- **Memory**. The Memory tasks assess a model's ability to retain and recall visual object information across irrelevant image inputs. Specifically, each trial contains one or more object frames followed by blank or distractor frames. The agent must encode the target's properties on the initial frame and accurately report them when prompted after the interruption.

For each type of task, there are two task variants: Report (R) and Compare (C). These refer to whether the task requires the agent to report an object property or requires the agent to compare two objects by their properties. The type of property which is required to report or compare is either object location (Loc) or object category (Cat). See Figure 1 and Appendix Figures A.1.1-A.1.5 for example task trials.

To probe how VLMs' PAM scores relate to their more general visual reasoning capabilities, we additionally evaluate each model on a set of composite visual reasoning tasks (CVR) that each involve various combinations of the different cognitive abilities. The CVR tasks were randomly generated using the iWISDM `AutoTask` framework. The three levels of complexity, Low (L), Medium (M), and High (H), were generated following the `AutoTask` parameters outlined in Lei et al. [22] and can be found in Appendix Table A.4.5. Altogether, we tested each model on 22 vision-language tasks (see Appendix Table A.4.3 for the full list).

### 3.3 Vision Language Models

We tested seven different vision language models: InternVL2.5-8B [55], LLaVa-OneVision-7B [56], MiniCPM-V-2.6-8B [57], Qwen2.5-VL-7B [14], GPT-4o-Mini, and GPT-4o [10]. These models were chosen as a representative state-of-the-art set of open-source (MiniCPM-V, InternVL2.5, LLaVa-OneVision, and Qwen2.5-VL) and proprietary (GPT-4o and GPT-4o-Mini) model series. We focused on evaluating smaller open-source models, as they often achieve performance comparable to their larger counterparts while offering greater computational efficiency and broader practical usability. However, to gain more insight into whether our evaluations on open-source models revealed properties that were due to their smaller size rather than other factors, we performed further tests on Qwen2.5-VL-72B, the largest size of this model.

GPT-4o and GPT-4o-Mini were evaluated with the official API. All variations of Qwen2.5-VL and LLaVa-OneVision were hosted with llama-factory [58] and evaluated with the OpenAI-style API. The InternVL-2.5 and MiniCPM-V 2.6 were deployed using Hugging Face Transformers. All models were evaluated with a (near-)identical task prompt template: the prompt describes the tasks and

possible answers, includes the trial's instructions and images, and suggests Chain-of-Thought reasoning (see Appendix Figure A.11.1 for the full prompt and OpenAI API code). After the VLMs are run on the benchmark task, the responses are passed to a Qwen2.5-72B LLM, which is prompted to extract the final answers given a list of possible answers. From these final answers, accuracy scores are calculated for all benchmarks. The GPT-4o-Mini and GPT-4o results were collected using gpt-4o-mini-2024-07-18 and gpt-4o-2024-08-06 snapshots provided by OpenAI.

### 3.4 Decoupling Vision & Text via Captioning

Aside from GPT-4o, our findings show notable disparities in cognitive performance between open-source models and humans on visual reasoning tasks. To address this, we performed a set of experiments designed to enhance the core cognitive and visual reasoning abilities of Qwen2.5-VL-7B via vision-text decoupling. Implemented through prompt modifications, we vary the required processing of visual information as well as reasoning load through three methods:

- **PC (Pre-captioned):** All images in the task prompt are replaced with ground truth captions containing both category and location object information. This method allows us to confirm whether any visual reasoning weaknesses on our datasets result from the LLM not being able to sufficiently understand the semantics of the prompt instructions and questions.[2]

- **SC (Self-captioning):** In separate conversations, models are instructed to caption an image with information on object category and location. These model *self-captions* are then used to replace images within a given task prompt. This method is related to the decoupling method from [59]; however, unlike the Prism framework, our method does not rely on a separate LLM to perform the text-only reasoning.

- **SC-I (Self-captioning-Interleaved):** Instead of simply replacing images with model self-captions, we interleave the captions between the images such that all task images are followed by their corresponding self-caption.

Qwen2.5-VL-7B was selected as the primary VLM for these experiments due to its substantial performance gap relative to both GPT-4o and humans, while also holding its status as the most recent open-source model in our evaluation set. To ensure the generality of our findings, we also applied the key vision-text decoupling experiments to GPT-4o and to the larger 72B size of Qwen2.5-VL.

Example prompts and Python code for our captioning methods can be found in Appendix A.11.

## 4 Results

### 4.1 Cognitive Evaluation of VLMs with PAM Dataset

We first evaluated selected VLMs and humans (see Section A.2 for details) on the PAM dataset, split by Location and Category subtask types. The results listed in Table 1 present the average accuracies on each task group (see Appendix Table A.4.1 for more granular performance scores). Overall, GPT-4o performed best, with scores nearing human levels, while GPT-4o-Mini closely followed, matching human levels on category tasks. Between the four compact open-source models, performances were extremely dependent on the cognitive axis being probed. Both open-source and proprietary models showed the pre-established weakness for spatial-based visual reasoning, with open-source models and GPT-4o-Mini expressing large performance gaps between location and category task variants.

In addition to task accuracy, we recorded human response times to provide context on the perceived difficulty of these tasks (see Appendix Table A.2.1). The response times clearly correlate with task complexity; for instance, the average time for a high-complexity category CVR task (77.21s) was more than ten times longer than for a simple perception task (7.39s). Interestingly, humans consistently took longer to solve category-based tasks than their location-based counterparts, a pattern that contrasts with the performance of most VLMs, which struggle more with spatial reasoning. A difference likely reflecting the abundance of object category based task training data in comparison to spatial task data.

---

[2]The PC method's ground-truth captions are made from the iWISDM task trial meta-data.

Table 1: Average PAM and CVR scores across all tasks for various open-source models, closed-source models, and human performance. Scores are presented as mean percentage accuracy ($\pm$ standard deviation). Best performance for open-source models for each task is **bolded**. Best performance overall for each task is underlined.

| Task | LLaVa | MiniCPMV | InternVL | Qwen-7B | *4o-Mini* | *4o* | *Human* |
|---|---|---|---|---|---|---|---|
| Percep. (Cat) | $71.00^{\pm5.11}$ | $82.67^{\pm4.28}$ | $72.67^{\pm5.02}$ | $\mathbf{83.67}^{\pm4.18}$ | $89.00^{\pm3.55}$ | $90.33^{\pm3.36}$ | $\underline{93.75}^{\pm5.55}$ |
| Percep. (Loc) | $67.00^{\pm5.29}$ | $39.00^{\pm5.49}$ | $\mathbf{75.00}^{\pm4.88}$ | $44.33^{\pm5.59}$ | $57.67^{\pm5.56}$ | $89.00^{\pm3.55}$ | $\underline{97.50}^{\pm3.98}$ |
| Feature Attn. | $62.00^{\pm2.59}$ | $54.30^{\pm2.65}$ | $\mathbf{64.67}^{\pm2.55}$ | $55.85^{\pm2.65}$ | $71.85^{\pm2.40}$ | $87.19^{\pm1.78}$ | $\underline{98.75}^{\pm3.27}$ |
| Spatial Attn. | $54.52^{\pm2.65}$ | $58.67^{\pm2.62}$ | $\mathbf{60.89}^{\pm2.60}$ | $57.48^{\pm2.63}$ | $68.81^{\pm2.47}$ | $75.93^{\pm2.28}$ | $\underline{98.75}^{\pm3.27}$ |
| Memory (Cat) | $66.83^{\pm1.51}$ | $73.68^{\pm1.41}$ | $61.20^{\pm1.56}$ | $\mathbf{78.59}^{\pm1.31}$ | $88.96^{\pm1.00}$ | $91.47^{\pm0.89}$ | $\underline{96.88}^{\pm2.88}$ |
| Memory (Loc) | $\mathbf{50.83}^{\pm1.63}$ | $39.11^{\pm1.59}$ | $48.75^{\pm1.63}$ | $42.22^{\pm1.61}$ | $57.28^{\pm1.62}$ | $83.47^{\pm1.21}$ | $\underline{96.25}^{\pm3.10}$ |
| CVR-Cat-L | $46.00^{\pm7.88}$ | $\mathbf{62.00}^{\pm7.68}$ | $52.00^{\pm7.89}$ | $60.00^{\pm7.75}$ | $81.33^{\pm6.21}$ | $\underline{91.33}^{\pm4.56}$ | $82.50^{\pm11.60}$ |
| CVR-Loc-L | $\mathbf{66.00}^{\pm7.50}$ | $50.00^{\pm7.90}$ | $46.00^{\pm7.88}$ | $56.00^{\pm7.85}$ | $51.33^{\pm7.90}$ | $66.00^{\pm7.50}$ | $\underline{92.50}^{\pm8.64}$ |
| CVR-Cat-M | $44.00^{\pm7.85}$ | $49.33^{\pm7.90}$ | $38.00^{\pm7.68}$ | $\mathbf{54.00}^{\pm7.88}$ | $72.00^{\pm7.12}$ | $\underline{96.67}^{\pm3.07}$ | $95.00^{\pm7.56}$ |
| CVR-Loc-M | $58.00^{\pm7.82}$ | $\mathbf{59.33}^{\pm7.77}$ | $50.00^{\pm7.90}$ | $49.33^{\pm7.90}$ | $51.33^{\pm7.90}$ | $\underline{82.67}^{\pm6.04}$ | $70.00^{\pm13.68}$ |
| CVR-Cat-H | $24.00^{\pm4.81}$ | $37.00^{\pm5.43}$ | $37.33^{\pm5.44}$ | $\mathbf{39.33}^{\pm5.49}$ | $63.00^{\pm5.43}$ | $\underline{83.67}^{\pm4.18}$ | $72.50^{\pm13.36}$ |
| CVR-Loc-H | $20.00^{\pm4.51}$ | $31.00^{\pm5.21}$ | $\mathbf{36.33}^{\pm5.41}$ | $29.33^{\pm5.13}$ | $39.67^{\pm5.50}$ | $64.67^{\pm5.38}$ | $\underline{75.00}^{\pm13.00}$ |

### 4.1.1 VLM Perception

Consistent with prior literature, our perception tasks reveal that VLMs are much better at object categorization than they are at object localization. GPT-4o, representing the top proprietary VLMs, had the smallest gap between these subtask types. However, its miniature variant (GPT-4o-Mini), as well as all tested open-source models, showed significant weaknesses on tasks requiring the perception of spatial object information. Interestingly, taking a look at the granular results of Table A.4.1, we see significant variations in performance between Perception comparison (C) and report (R) task types. For example, LLaVA-OneVision-7B achieves 88% accuracy and exceeds humans on Perc-Loc-R, but achieves only 46% on Perc-Loc-C. While an opposite bias can be seen for other models and for category variants. These results reveal that VLMs not only have pervasive perception limitations but also harbor granular biases that seem to vary greatly depending on specific task requirements and structure. The fact that many VLMs find report tasks more difficult than comparison tasks is surprising, and as shown below, these peculiar biases also appear to be prevalent in VLM Attention and Memory abilities.

### 4.1.2 VLM Attention

The Attention tasks proved more of a challenge for the proprietary models. While still the best model tested, here we observed GPT-4o's first major gap with human performance. Specifically, GPT-4o struggles with attending to task-relevant objects while ignoring irrelevant ones. This was especially true for spatial attention, where relevant objects are specified by their location. All open-source models performed far below human levels. We had expected the differences between Feature and Spatial Attention performances to be smaller than those between the Location and Category variants of the other task types. This is because both of these Attention task variants require simultaneous understanding of location and category. In contrast to GPT-4o, the compact models largely met that expectation of similar performance between Feature and Spatial Attention task variants.

### 4.1.3 VLM Memory

Finally, for Memory, most models perform similarly to their evaluation on Perception, suggesting a robustness to additional irrelevant frames. GPT-4o outperforms all models once again, reaching human-level accuracy on the Category variant. However, like Spatial Attention, location-based Memory tasks revealed another gap remaining between humans and GPT-4o. GPT-4o-Mini follows a similar pattern, with exceptional category-based Memory task performance and weak location-based Memory task performance. Similar biases are present for all other models. A granular look with Appendix Table A.4.1 also reveals GPT-4o struggles more with Memory comparison tasks when the delay frames contain irrelevant objects.

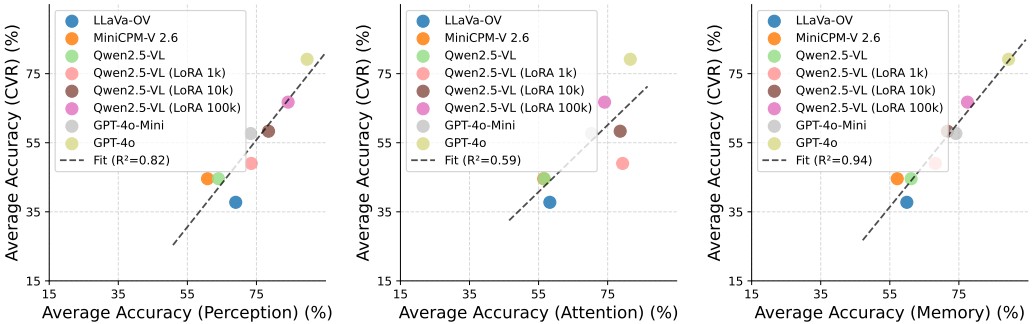

Figure 2: Scatter plots comparing average PAM task performance against average CVR task performance across all models. Each point represents a different model or Qwen2.5-VL-7B LoRA versions. The x-axis shows the average accuracy on the specified PAM task category (averaging Loc and Cat), and the y-axis shows the CVR accuracy score.

## 4.2 Evaluating VLMs on Composite Visual Reasoning

With their core cognitive abilities established, we turned to assessing how well the models perform composite visual reasoning (CVR) tasks. For this, we tested all models on the set of procedurally generated CVR tasks of varying levels of complexity (Table 1). We find that GPT-4o was consistently the strongest model across all complexities, even exceeding humans on most tasks. GPT-4o-Mini managed to perform near this level on the low-complexity category variant. However, a significant gap emerged in all other CVR tasks. The open-source models demonstrate pronounced drops in performance with increasing complexity of the tasks. Interestingly, GPT-4o seems to be better at solving CVR tasks that have Medium-level complexity than Low-level complexity. This difference could be due to a CoT bias towards *if-then-else* operations, which are excluded from the Low complexity task generation. However, a detailed analysis inspecting the outputs of GPT-4o across these two complexities would be needed to confirm this.

Relative to the PAM evaluation, GPT-4o performed much worse on the location-based variants compared to their category-based counterparts. This may stem from small performance differences in PAM tasks compounding within the more demanding CVR setting, leading to larger gaps. The result highlights the importance of addressing even minor biases in core cognitive abilities.

### 4.2.1 Relationship Between PAM & CVR

Intuitively, models with stronger memory, attention, and perception abilities should also perform better on tasks that require a combination of these skills. Since our CVR tasks are designed to engage exactly these abilities in combination, we plot each model's PAM performance against its CVR performance, as shown in Figure 2. All performances on all three core cognitive axes showed strong correlation to CVR task performance. Memory task performance stands out with a correlation of 0.94. This is likely due to the abundance of Memory comparison-based subtasks (i.e comparing two objects with irrelevant images in between), contained within CVR tasks. Furthermore, models' PAM and CVR accuracies are significantly correlated to their performance on widely used benchmarks such as MMMU-Pro (Figures A.7.1), which further validates the effectiveness of the evaluations presented here.

## 4.3 What It Takes for a VLM to Reason

Our results reveal substantial gaps in cognitive performance between open-source models and humans on visual reasoning tasks. In this section, we explore ways to enhance the core reasoning abilities of VLMs. We start by testing a series of prompt modifications to assess how different levels of vision-text decoupling affect Qwen2.5-VL-7B. These experiments introduce a simple yet effective strategy to boost visual reasoning while also highlighting key architectural bottlenecks. We then evaluate the impact of supervised LoRA fine-tuning on Qwen2.5-VL-7B using a separate set of CVR tasks.

### 4.3.1 Decoupling Vision & Text via Captioning

Table 2: PAM and CVR performance of Qwen2.5-VL-7B using different captioning methods. The Base column shows absolute percentage accuracies ($\pm$ standard deviation). Subsequent columns show the percentage change in accuracy from the Base method ($\pm$ standard deviation). Positive changes are shown in green, and negative changes in red. Asterisks denote statistical significance: * $p < 0.05$, ** $p < 0.01$, *** $p < 0.001$.

| Task | Base | SC | SC-I | PC |
|---|---|---|---|---|
| Perception (Cat) | $83.67^{\pm4.18}$ | $+1.33^{\pm4.04}$ | $-6.17^{\pm5.76}$ | $-2.00^{\pm4.37}$ |
| Perception (Loc) | $44.33^{\pm5.59}$ | $+28.67*^{\pm5.00}$ | $+10.17^{\pm6.84}$ | $+55.00*^{\pm1.11}$ |
| Feature Attention | $55.85^{\pm2.65}$ | $+22.00***^{\pm2.21}$ | $-1.74^{\pm3.25}$ | $+22.22***^{\pm2.21}$ |
| Spatial Attention | $57.48^{\pm2.63}$ | $+18.15***^{\pm2.29}$ | $-1.70^{\pm3.24}$ | $+8.52^{\pm2.52}$ |
| Memory (Cat) | $78.59^{\pm1.31}$ | $+4.72***^{\pm1.19}$ | $-5.27***^{\pm1.73}$ | $+6.27***^{\pm1.15}$ |
| Memory (Loc) | $42.22^{\pm1.61}$ | $+26.89***^{\pm1.51}$ | $+17.07***^{\pm1.96}$ | $+47.39***^{\pm1.00}$ |
| CVR-Cat-L | $60.00^{\pm7.75}$ | $+2.67^{\pm7.65}$ | $+1.00^{\pm9.39}$ | $+1.33^{\pm7.70}$ |
| CVR-Loc-L | $56.00^{\pm7.85}$ | $+8.67^{\pm7.56}$ | $+0.00^{\pm9.55}$ | $+12.67^{\pm7.34}$ |
| CVR-Cat-M | $54.00^{\pm7.88}$ | $+3.33^{\pm7.82}$ | $+7.00^{\pm9.39}$ | $+12.67^{\pm7.46}$ |
| CVR-Loc-M | $49.33^{\pm7.90}$ | $+10.00^{\pm7.77}$ | $+2.67^{\pm9.61}$ | $+10.67^{\pm7.75}$ |
| CVR-Cat-H | $39.33^{\pm5.49}$ | $+6.00^{\pm5.60}$ | $+6.67^{\pm6.84}$ | $+13.00^{\pm5.62}$ |
| CVR-Loc-H | $29.33^{\pm5.13}$ | $+17.33*^{\pm5.61}$ | $+7.17^{\pm6.61}$ | $+32.00*^{\pm5.48}$ |

Considering the well-established reasoning abilities of LLMs, we used self-captioning and pre-captioning methods to investigate the source of poor visual reasoning in VLMs through vision-text decoupling. The results for these experiments are displayed in Table 2; for a fully granular look, please see Appendix Table A.4.2. We also evaluated one-shot prompting, however this approach did not yield a significant performance boost compared to other methods, a finding that aligns well with previous literature [60, 61].

Our pre-captioning method led to the largest and most consistent performance gains across the PAM tasks, with these performance gains also reflected in CVR tasks. Most importantly, the SC method also led to significant improvements on all task sets across PAM and CVR datasets.

The PC results suggest that the underlying language model components of Qwen2.5-VL-7B are capable of interpreting the semantics of cognitive task instructions and performing the required reasoning. Replacing images with self-generated captions (SC) of those same images proved to be a simple yet effective strategy for enhancing Qwen2.5-VL-7B's cognitive performance. The comparable performance between PC and SC indicates that the model can caption at least single images with sufficient accuracy to support PAM and CVR tasks. Therefore, any performance differences between PC and SC are likely attributable to the core language model's ability to translate visual tokens into accurate captions.

Notably, the largest performance gains by far for the SC method compared to the baseline were for location tasks. Across both PAM and CVR results, applying SC to location-only tasks saw an average accuracy improvement of 18.31% compared to 3.54% for category tasks. This stark difference suggests that the bottleneck of VLM spatial reasoning may stem from inadequate CoT training. Two key observations support this: (1) Qwen2.5-VL-7B can accurately describe object locations from visual inputs, and use those descriptions to answer location-based Perception questions, and (2) Qwen2.5-VL-7B fails at the same questions when no text-based location information is provided. These simple observations point to a core limitation in current CoT strategies, even for single-image spatial reasoning, and likely explain the near-chance performance on multi-image CVR and PAM location tasks. By contrast, it is unclear why this bottleneck affects category tasks less. A likely reason is the abundance of object recognition data in VLM training.

To assess the broader applicability of our self-captioning (SC) method for improving VLM performance, we conducted additional experiments on GPT-4o and the largest variant of Qwen2.5-VL. The results, presented in Table A.5.1 and A.5.2, demonstrate that the effectiveness of the SC method extends beyond Qwen2.5-VL-7B. These findings further confirm our CoT diagnosis, showing that comparable improvements can be gained in larger and SOTA proprietary VLMs.

Finally, we used the SC-I method on Qwen2.5-VL-7B to test the effect of including images in the model's input on its performance. In this setup, the model is provided with its self-caption descriptions interleaved between the corresponding images. This method saw significant improvements on location-only PAM tasks and almost all CVR tasks. However, on category-only PAM tasks, the addition of images decreased performance relative to the baseline. Furthermore, SC-I improvements were significantly smaller than those of the image-free SC method. These results suggest the inclusion of images alone leads to diminished reasoning abilities, likely resulting from attention capacity (see Attention Analysis in Appendix A.9) or interference issues. These results align with recent mechanistic interpretability findings from [28] that show attention limitations are a contributing factor to spatial reasoning weaknesses in VLMs.

### 4.3.2 Fine-tuning VLMs

Table 3: PAM and CVR performance of Qwen2.5-VL-7B base model and fine-tuned variants using LoRA with different amounts of training data. The Base column shows absolute percentage accuracies ($\pm$ standard deviation). Subsequent columns show the percentage change in accuracy from the Base method ($\pm$ standard deviation). Positive changes are shown in green, and negative in red. Asterisks denote statistical significance: * $p < 0.05$, ** $p < 0.01$, *** $p < 0.001$.

| Task | Base | LoRA 1k | LoRA 10k | LoRA 100k |
|---|---|---|---|---|
| Perception (Cat) | $83.67^{\pm 4.18}$ | $+2.00^{\pm 3.97}$ | $+2.67^{\pm 3.89}$ | $+6.00*^{\pm 3.46}$ |
| Perception (Loc) | $44.33^{\pm 5.59}$ | $+17.00^{\pm 5.48}$ | $+26.33^{\pm 5.13}$ | $+34.33*^{\pm 4.62}$ |
| Feature Attention | $55.85^{\pm 2.65}$ | $+20.74***^{\pm 2.26}$ | $+21.41***^{\pm 2.23}$ | $+16.96***^{\pm 2.37}$ |
| Spatial Attention | $57.48^{\pm 2.63}$ | $+24.59***^{\pm 2.05}$ | $+22.52***^{\pm 2.13}$ | $+17.93***^{\pm 2.30}$ |
| Memory (Cat) | $78.59^{\pm 1.31}$ | $+2.35**^{\pm 1.26}$ | $-3.20*^{\pm 1.38}$ | $+3.68**^{\pm 1.22}$ |
| Memory (Loc) | $42.22^{\pm 1.61}$ | $+10.78***^{\pm 1.63}$ | $+22.72***^{\pm 1.56}$ | $+27.69***^{\pm 1.50}$ |
| CVR-Cat-L | $60.00^{\pm 7.75}$ | $-11.33^{\pm 7.90}$ | $-9.33^{\pm 7.90}$ | $+8.00^{\pm 7.38}$ |
| CVR-Loc-L | $56.00^{\pm 7.85}$ | $+0.00^{\pm 7.85}$ | $-5.33^{\pm 7.90}$ | $+10.00^{\pm 7.50}$ |
| CVR-Cat-M | $54.00^{\pm 7.88}$ | $+1.33^{\pm 7.86}$ | $+6.67^{\pm 7.72}$ | $+8.67^{\pm 7.65}$ |
| CVR-Loc-M | $49.33^{\pm 7.90}$ | $+7.33^{\pm 7.83}$ | $-8.67^{\pm 7.77}$ | $+4.67^{\pm 7.88}$ |
| CVR-Cat-H | $39.33^{\pm 5.49}$ | $+9.33*^{\pm 5.62}$ | $+26.67*^{\pm 5.33}$ | $+28.67*^{\pm 5.25}$ |
| CVR-Loc-H | $29.33^{\pm 5.13}$ | $+9.67^{\pm 5.49}$ | $+36.67*^{\pm 5.33}$ | $+44.33*^{\pm 4.96}$ |

We investigated whether fine-tuning Qwen2.5-VL-7B on random CVR tasks leads to improvements on the core cognitive tasks of the PAM dataset (*Perception, Attention, and Memory*; Table 3). To prove the generalization of the fine-tuning, we also performed LoRA fine-tuning on Qwen2.5-VL-32B model on the same data (see Table A.6.1). We used supervised fine-tuning with LoRA on randomly generated CVR tasks (please see Section A.3 for method details). We found that fine-tuning on even 1,000 trials yields large gains in performance on core cognitive tasks, and the performance gains further increase with additional data. Similar to our method of self-captioning, fine-tuning appears to disproportionately improve location-based PAM task performance. Unsurprisingly, training on 100,000 held-out CVR task trials improved CVR task performance the most.

While these results are promising, standard LoRA methods are known to be prone to overfitting [62–64]. To mitigate this risk and make sure no severe overfitting has occurred in the Qwen2.5-VL-7B model during fine-tuning, we experimented with several dropout rates. Our training runs with lower dropout ratios (0.0 and 0.1) experienced overfitting, resulting in a rise in the validation loss during training. Consequently, we selected a dropout rate of 0.2 for our final models. To further assess generalization, we benchmark the LoRA fine-tuned models along with the base model on MMBench, MMMU-Pro and VQAv2. Table A.3.2, A.3.3 shows that supervised LoRA fine-tuning on CVR tasks can provide modest improvements to MMBench and VQAv2. However, performance on MMMU-Pro slightly decreases when fine-tuned on smaller dataset sizes.

## 5 Discussion

This work provides a detailed analysis of the cognitive capabilities of modern VLMs by systematically evaluating them along the core axes of Perception, Attention, and Memory, using the PAM

dataset, as well as on combinations of these abilities via the CVR dataset. Our findings reveal distinct cognitive profiles: SOTA models like GPT-4o demonstrate strong perceptual abilities for object categories and reasonable memory, but exhibit significant weaknesses in processing spatial information and performing comparative judgments across frames or objects. The spatial reasoning deficit in particular is a pervasive issue across most VLMs tested.

Our analyses strongly suggest that a primary bottleneck lies not in the fundamental reasoning capacity of the underlying LLM, nor purely in low-level visual perception, but rather in the effective *integration* of visual features into the reasoning process. Models often possess the visual information (as shown by self-captioning success) but struggle to utilize it correctly when solving tasks directly from images. A limitation that a sufficiently robust CoT strategy should be able to overcome. These same analyses provided a simple vision-text decoupling method for which this bottleneck can be easily mitigated. We demonstrate that this method leads to consistent visual reasoning performance gains, even for a SOTA VLM that already rivals human-level performance.

Fine-tuning VLMs on a small dataset of random visual reasoning tasks also led to substantial improvements along all cognitive axes while yielding slight gains on out-of-distribution benchmarks. We believe it is likely that more sophisticated methods of fine-tuning, such as Reinforcement Learning with CoT reward [65], could lead to improved generalization. This hypothesis is supported by high correlations when comparing model PAM and CVR accuracy against established benchmarks like MMMU-Pro and VQAv2, which can be found in Figures A.7.1 and A.8.1. This is an important finding that further validates our evaluation datasets and approach.

## 6    Limitations

Our study has several limitations that future work should address: 1) The stimulus set used to generate test samples consisted of synthetic objects from only eight categories of everyday items. Future evaluations could incorporate a wider variety of objects, including more diverse and natural visual categories. 2) Visual frames in our benchmarks featured uniformly colored backgrounds, consistent with practices in cognitive science and neuroscience. However, future studies could adopt more naturalistic settings where objects are embedded in complex scenes. Leveraging generative image models may offer a promising avenue for creating such stimuli [66–68]. 3) For feasibility, we restricted our fine-tuning experiments to the Qwen2.5-VL model and used up to 100k trials. Expanding these experiments to larger models and datasets could yield additional insights. Generative task environments, such as those proposed by Lei et al. [22], provide a convenient framework for scaling data to arbitrarily large sizes. 4) Our fine-tuning experiments consisted of tasks with a random level of complexity. While fine-tuning on these tasks already resulted in improvements on the cognitive benchmarks, we expect that a more balanced data distribution would be even more helpful. Future studies could perform fine-tuning on more complexity-balanced task datasets. 5) The self-captioning (SC) method, while effective at improving performance, introduces a computational trade-off. Our analysis shows that this method has roughly double the inference runtime compared to the standard direct-input approach, which could be a critical factor in real-world applications. A breakdown of this analysis is provided in Appendix A.10.

## Acknowledgments

This research was supported by the Healthy-Brains-Healthy-Lives startup supplement grant, the NSERC Discovery grant RGPIN-2021-03035, and CIHR Project Grant PJT-191957. P.B. was supported by FRQ-S Research Scholars Junior 1 grant 310924, and the William Dawson Scholar award. M.P. was supported by the UNIQUE Masters and PhD Fellowships, and the Stichting Formation Award. All analyses were executed using resources provided by the Digital Research Alliance of Canada (Compute Canada) and funding from Canada Foundation for Innovation project number 42730. Z.W. was supported by China Scholarship Council Grant 202506070003. L.G was supported by the UNIQUE MSc Excellence Scholarship 2024-2025.

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

# A Appendix

## A.1 Task Examples

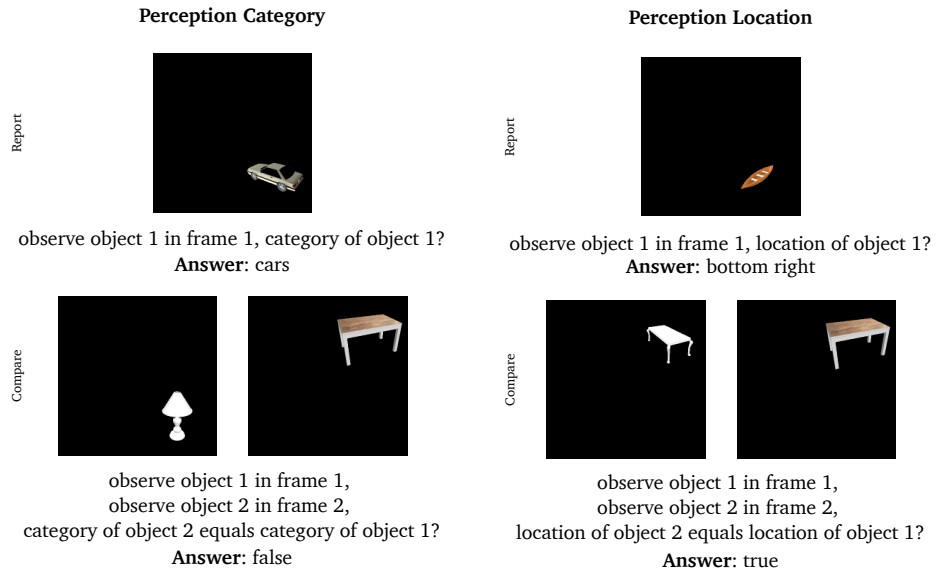

Figure A.1.1: Example trials from Perception Category (Perc-Cat-R & Perc-Cat-C) and Localization (Perc-Loc-R & Perc-Loc-C) tasks. Each task consists of two variations: *Report* where the agent is tasked with reporting the object's property; and *Compare* where the agent is tasked with comparing that property between two objects on separate frames.

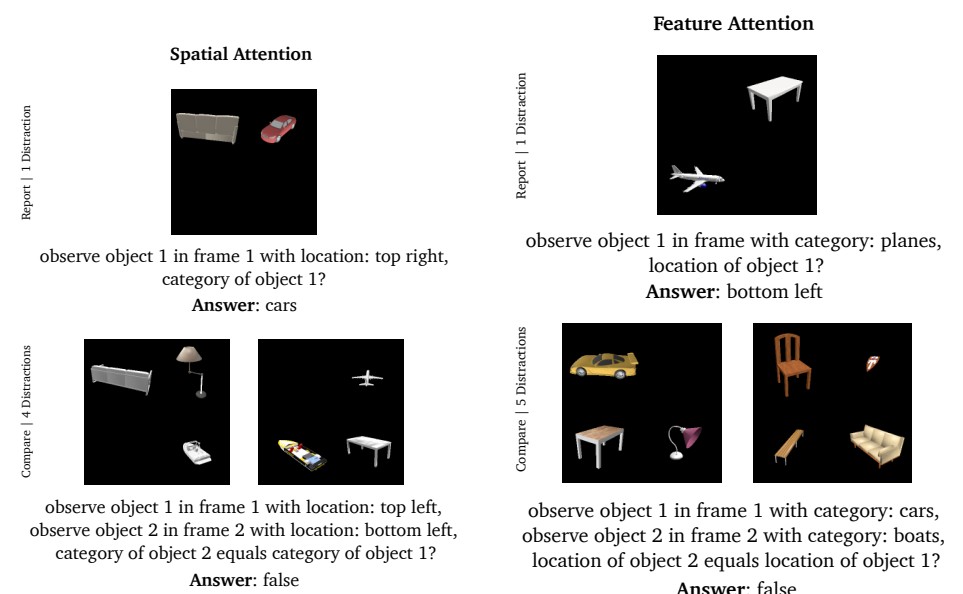

Figure A.1.2: Example trials from Spatial (Att-Spa-R & Att-Spa-C) and Feature Attention (Att-Feat-R & Att-Feat-C) tasks. Each task consists of two variations: *Report*, where the agent is tasked with reporting the object's property, and *Compare*, where the agent is tasked with comparing that property between two objects on separate frames.

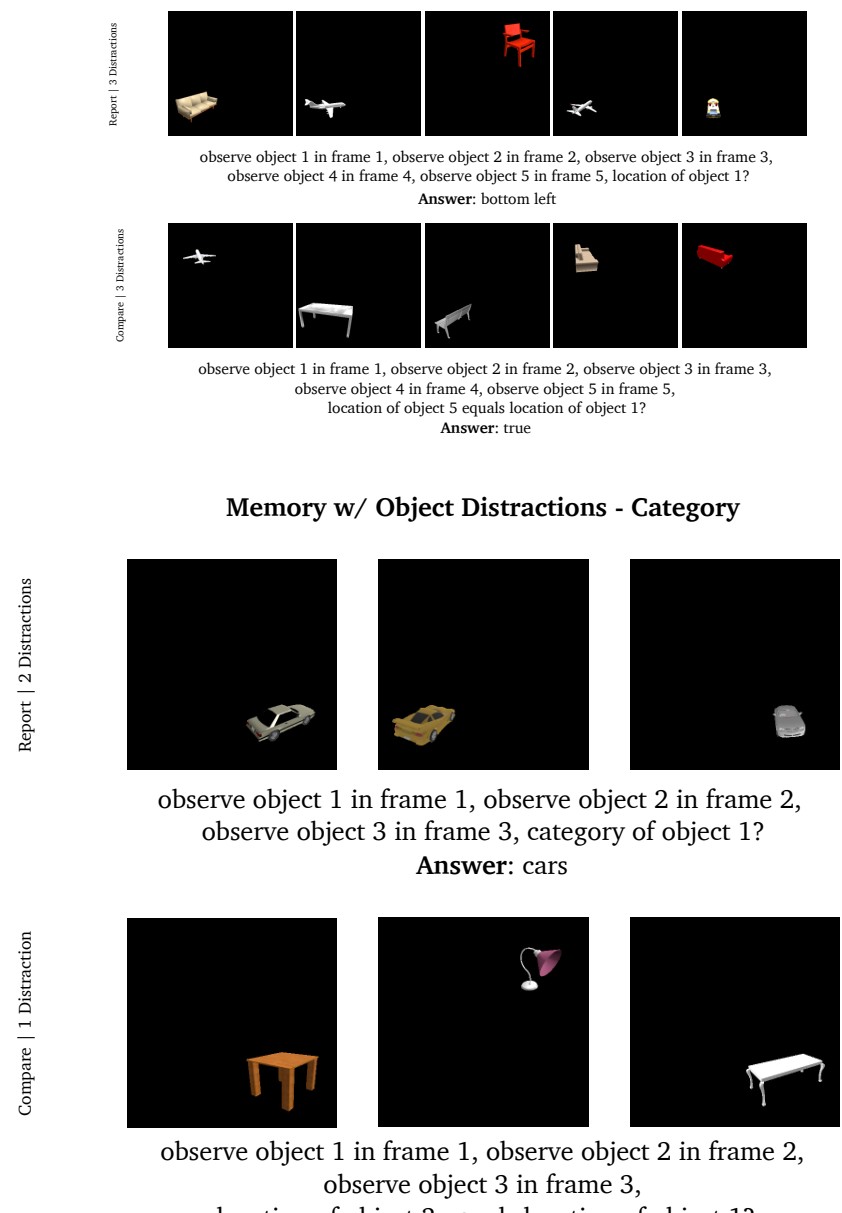

Figure A.1.3: Example trials from Memory with Distractors Category (Mem-Dis-Cat-R & Mem-Dis-Cat-C) and Memory with Distractors Location (Mem-Dis-Loc-R & Mem-Dis-Loc-C) Memory tasks. Each task consists of two variations: *Report*, where the agent is tasked with reporting the object's property, and *Compare*, where the agent is tasked with comparing that property between two objects on separate frames.

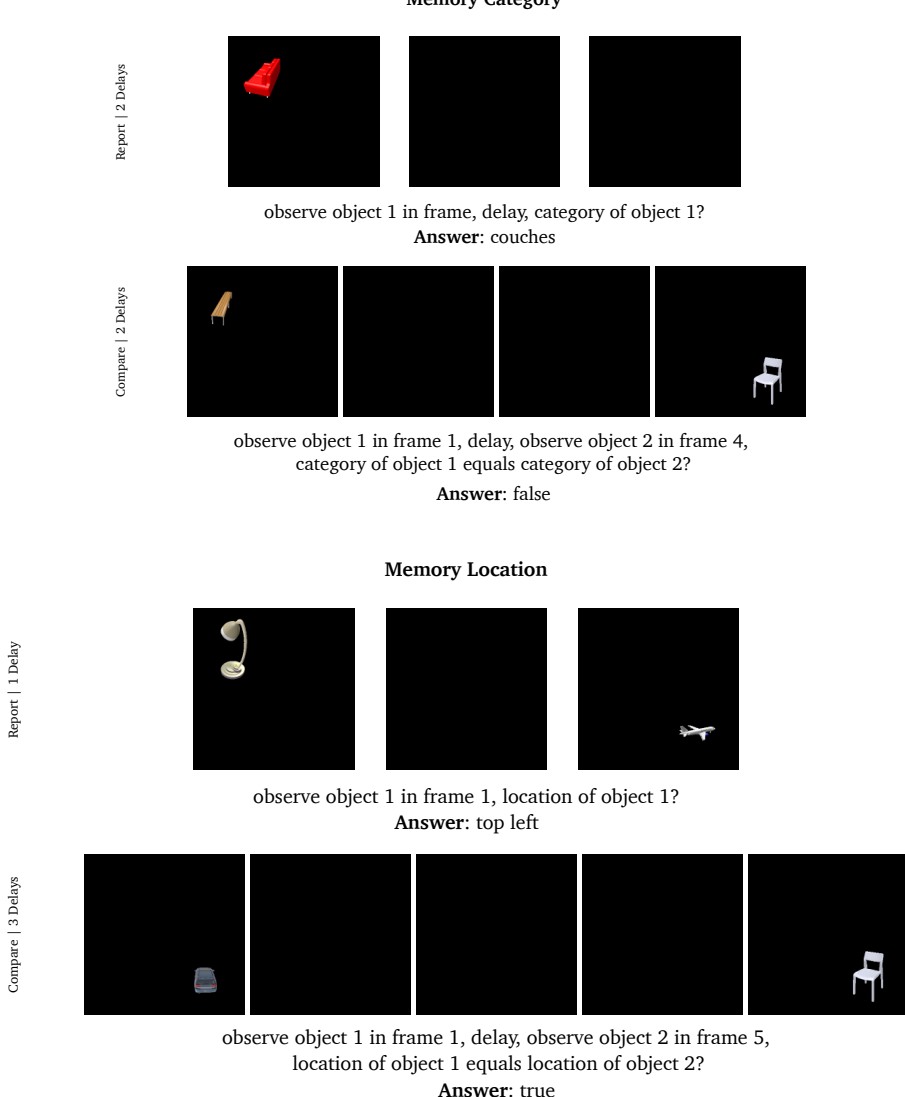

Figure A.1.4: Example trials from Memory Category (Mem-Cat-R & Mem-Cat-C) and Location (Mem-Loc-R & Mem-Loc-C) tasks. Each task consists of two variations: *Report* where the agent is tasked with reporting the object's property, and *Compare* where the agent is tasked with comparing that property between two objects on separate frames.

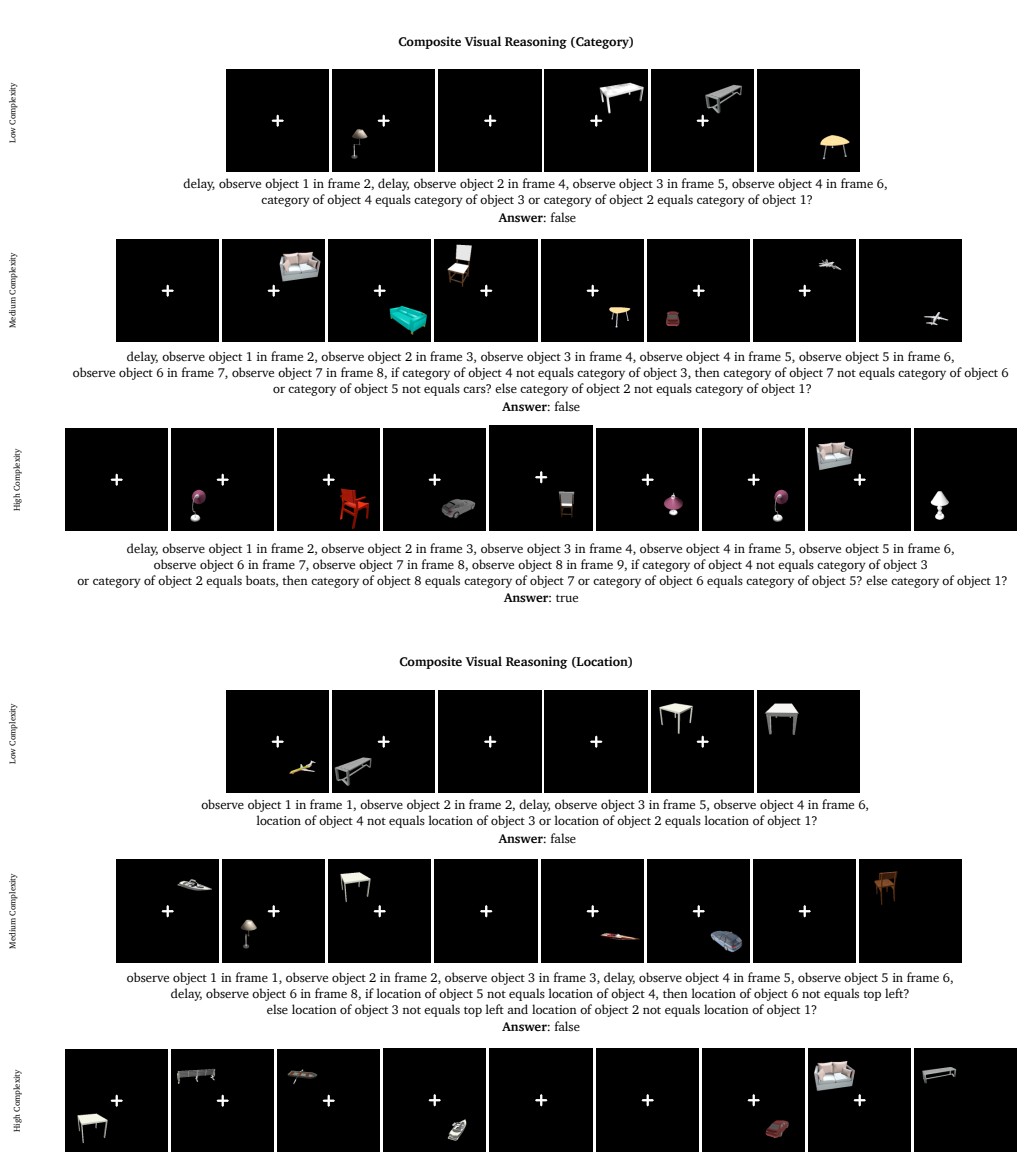

Figure A.1.5: Example trials from Composite Visual Reasoning Category (CVR-Cat-H/M/L) and Location (CVR-Loc-H/M/L) tasks. These tasks were randomly generated into three sets of varying complexities: High (H), Medium (M), and Low (L).

## A.2 Human Baseline Details

To establish human baselines, eight people were tasked with performing all 22 different task types across the PAM and CVR datasets. Each subject solved 5 trials of every task, providing a total of 880 human accuracy data points. No compensation was given to subjects.

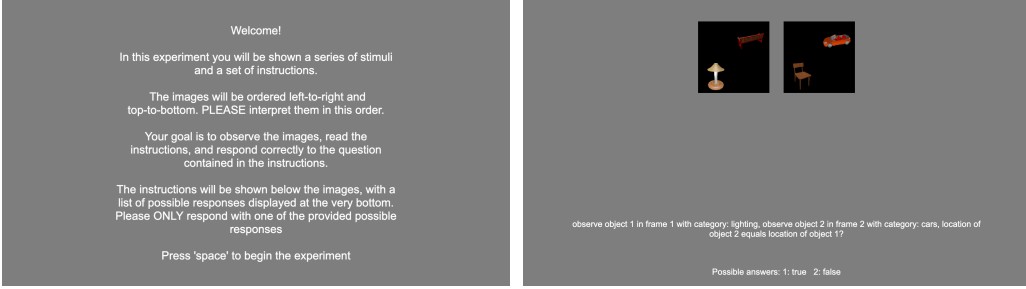

Figure A.2.1: Screenshots of human subject instructions (left) and a task trial example (right).

Table A.2.1: Average human response time across all task types.

| Task | Mean Response Time (s) |
|------|------------------------|
| Perception (Cat) | $7.39^{\pm 1.56}$ |
| Perception (Loc) | $5.77^{\pm 1.12}$ |
| Feature Attention | $11.34^{\pm 1.78}$ |
| Spatial Attention | $10.79^{\pm 1.89}$ |
| Memory (Cat) | $9.67^{\pm 1.30}$ |
| Memory (Loc) | $8.15^{\pm 1.05}$ |
| CVR-Cat-L | $30.68^{\pm 6.47}$ |
| CVR-Loc-L | $23.39^{\pm 6.83}$ |
| CVR-Cat-M | $43.81^{\pm 17.41}$ |
| CVR-Loc-M | $34.36^{\pm 9.74}$ |
| CVR-Cat-H | $77.21^{\pm 25.90}$ |
| CVR-Loc-H | $41.44^{\pm 10.84}$ |

## A.3 Supervised VLM Fine-tuning on Composite Visual Reasoning

As an additional attempt to improve visual reasoning, we performed fine-tuning experiments on Qwen2.5-VL-7B. The model was fine-tuned with a supervised objective to output only the correct final answer. This meant the model would likely learn to solve the tasks without the use of any CoT. This presents an interesting contrast to our captioning experiments. Qwen2.5-VL-7B was selected as the VLM for these experiments for the same reasons provided in Section 3.4.

To achieve this, we used iWISDM [22] to generate task sets of varying sizes: 100, 1,000, and 10,000 tasks. For each task, 10 trials were generated, resulting in training sets comprising 1,000, 10,000, and 100,000 trials, respectively.

The CVR tasks generated for LoRA fine-tuning have similar AutoTask [22] generation parameters to those used for the CVR evaluation tasks. However, while the distributions should be similar, the fine-tuning tasks have a lower bound of complexity to avoid overfitting to complex instructions. It is also important to note that there is no overlap between the training and CVR evaluation datasets.

We chose to keep the ViT vision encoder of Qwen2.5-VL-7B frozen during fine-tuning. This decision was based on prior studies showing that the vision encoder and vision projection components already capture sufficient visual information [48]. However, the vision projector was unfrozen during fine-tuning as it offers a lightweight yet potentially impactful set of vision-text alignment weights to adapt. As such, the nonlinear MLP vision token projection in Qwen2.5-VL-7B was unfrozen dur-

ing training along with the core LLM. All fine-tuning experiments were performed on 4 NVIDIA A5000 GPUs. Please see Appendix Table A.3.4 for training hyperparameter details.

Table A.3.1: The iWISDM `AutoTask` parameters used to generate the fine-tuning task sets.

| # of allowed and/or operators in task | # of switch operators | # of trial frames | root operators | boolean operators |
|---|---|---|---|---|
| 0-2 | 0-1 | 3-9 | IsSame, And, Or, NotSame, GetLoc, GetCategory | IsSame, And, Or, NotSame |

Table A.3.2: MMBench & MMMU-Pro benchmarking scores (%) before and after LoRA fine-tuning.

| N_tasks | 0 (base) | 100 | 1k | 10k |
|---|---|---|---|---|
| N_trials | 0 (base) | 1000 | 10k | 100k |
| **MMBench** | 82.85 | 83.41 | 83.63 | 83.30 |
| **MMMU-Pro** | 34.36 | 33.70 | 33.83 | 34.94 |

Table A.3.3: Benchmarking scores for Qwen2.5-VL-7B and Qwen2.5-VL-72B on the VQAv2 dataset with different caption techniques and finetuning configurations.

| | **7B Models** | | | | | **72B Models** | |
|---|---|---|---|---|---|---|---|
| | Base | SC | LoRA 1k | LoRA 10k | LoRA 100k | Base | SC |
| **Accuracy (%)** | 60.07 | 65.07 | 63.96 | 64.44 | 65.84 | 71.09 | 72.02 |

Table A.3.4: Overview of the hyperparameters for Qwen2.5-VL-7B-Instruct LoRA Fine-tuning.

| N_tasks | 100 | 1000 | 10000 |
|---|---|---|---|
| N_trials | 1000 | 10000 | 100000 |
| **N_epochs** | | 10 | |
| **Batch_size** | | 1 | |
| **Gradient_accum** | | 32 | |
| **Scheduler** | | Cosine | |
| **Peak_LR** | | 4e-05 | |
| **Warmup** | 1 | 7 | 70 |
| **Mixed_precision** | | bf16 | |
| **Optimizer** | | AdamW(0.01) | |
| **LoRA_rank** | | 8 | |
| **LoRA_alpha** | | 16 | |
| **LoRA_dropout** | | 0.2 | |
| **LoRA_targets** | | vision projector, LLM | |

## A.4 Granular Results & Details

Table A.4.1: Average granular accuracy (%) comparison across all tasks and models. Scores are presented as mean percentage accuracy (± standard deviation). LLaVa: LLaVa-OneVision-7B; MiniCPM: MiniCPM-V-2.6-8B; InternVL: InternVL2.5-8B; Qwen 7B: Qwen2.5-VL-7B; Qwen 72B: Qwen2.5-VL-72B; 4o-Mini: GPT-4o-Mini; 4o: GPT-4o; human: Human subjects.

| Task | LLaVa | MiniCPM | InternVL | Qwen 7B | Qwen 72B | 4o-Mini | 4o | human |
|---|---|---|---|---|---|---|---|---|
| Mem-Cat-R | $82.40^{\pm2.72}$ | $68.40^{\pm3.32}$ | $62.40^{\pm3.46}$ | $76.53^{\pm3.03}$ | $82.20^{\pm3.35}$ | $87.47^{\pm2.37}$ | $87.87^{\pm2.34}$ | $100.00^{\pm4.38}$ |
| Mem-Cat-C | $52.93^{\pm3.56}$ | $78.27^{\pm2.95}$ | $65.20^{\pm3.40}$ | $82.13^{\pm2.74}$ | $92.00^{\pm2.39}$ | $98.80^{\pm0.82}$ | $99.60^{\pm0.52}$ | $97.50^{\pm6.22}$ |
| Mem-Loc-R | $56.80^{\pm3.54}$ | $25.87^{\pm3.13}$ | $54.80^{\pm3.55}$ | $33.20^{\pm3.36}$ | $69.80^{\pm4.01}$ | $76.13^{\pm3.05}$ | $96.40^{\pm1.35}$ | $95.00^{\pm7.56}$ |
| Mem-Loc-C | $47.60^{\pm3.57}$ | $49.07^{\pm3.57}$ | $59.60^{\pm3.50}$ | $52.13^{\pm3.57}$ | $62.20^{\pm4.23}$ | $52.40^{\pm3.57}$ | $87.33^{\pm2.38}$ | $100.00^{\pm4.38}$ |
| Perc-Cat-R | $82.00^{\pm6.12}$ | $74.67^{\pm6.90}$ | $74.00^{\pm6.96}$ | $74.67^{\pm6.90}$ | $81.00^{\pm7.63}$ | $82.00^{\pm6.12}$ | $80.67^{\pm6.29}$ | $97.50^{\pm6.22}$ |
| Perc-Cat-C | $60.00^{\pm7.75}$ | $90.67^{\pm4.71}$ | $71.33^{\pm7.17}$ | $92.67^{\pm4.25}$ | $97.00^{\pm3.71}$ | $96.00^{\pm3.30}$ | $100.00^{\pm1.25}$ | $90.00^{\pm9.55}$ |
| Perc-Loc-R | $88.00^{\pm5.22}$ | $30.67^{\pm7.30}$ | $76.00^{\pm6.78}$ | $33.33^{\pm7.46}$ | $64.00^{\pm9.25}$ | $66.00^{\pm7.50}$ | $93.33^{\pm4.09}$ | $95.00^{\pm7.56}$ |
| Perc-Loc-C | $46.00^{\pm7.88}$ | $47.33^{\pm7.89}$ | $74.00^{\pm6.96}$ | $55.33^{\pm7.86}$ | $77.00^{\pm8.16}$ | $49.33^{\pm7.90}$ | $84.67^{\pm5.76}$ | $100.00^{\pm4.38}$ |
| CVR-Cat-H | $24.00^{\pm4.81}$ | $37.00^{\pm5.43}$ | $37.33^{\pm5.44}$ | $39.33^{\pm5.49}$ | $69.83^{\pm6.33}$ | $63.00^{\pm5.43}$ | $83.67^{\pm4.18}$ | $72.50^{\pm13.36}$ |
| CVR-Loc-H | $20.00^{\pm4.51}$ | $31.00^{\pm5.21}$ | $36.33^{\pm5.41}$ | $29.33^{\pm5.13}$ | $40.97^{\pm6.74}$ | $39.67^{\pm5.50}$ | $64.67^{\pm5.38}$ | $75.00^{\pm13.00}$ |
| CVR-Cat-M | $44.00^{\pm7.85}$ | $49.33^{\pm7.90}$ | $38.00^{\pm7.68}$ | $54.00^{\pm7.88}$ | $88.00^{\pm6.41}$ | $72.00^{\pm7.12}$ | $96.67^{\pm3.07}$ | $95.00^{\pm7.56}$ |
| CVR-Loc-M | $58.00^{\pm7.82}$ | $59.33^{\pm7.77}$ | $50.00^{\pm7.90}$ | $49.33^{\pm7.90}$ | $48.00^{\pm9.61}$ | $51.33^{\pm7.90}$ | $82.67^{\pm6.04}$ | $70.00^{\pm13.68}$ |
| CVR-Cat-L | $46.00^{\pm7.88}$ | $62.00^{\pm7.68}$ | $52.00^{\pm7.89}$ | $60.00^{\pm7.75}$ | $80.00^{\pm7.77}$ | $81.33^{\pm6.21}$ | $91.33^{\pm4.56}$ | $82.50^{\pm11.60}$ |
| CVR-Loc-L | $66.00^{\pm7.50}$ | $50.00^{\pm7.90}$ | $46.00^{\pm7.88}$ | $56.00^{\pm7.85}$ | $52.00^{\pm9.61}$ | $51.33^{\pm7.90}$ | $66.00^{\pm7.50}$ | $92.50^{\pm8.64}$ |
| Att-Feat-R | $88.00^{\pm3.01}$ | $61.11^{\pm4.49}$ | $78.67^{\pm3.78}$ | $58.67^{\pm4.53}$ | $85.67^{\pm3.97}$ | $83.56^{\pm3.42}$ | $96.22^{\pm1.80}$ | $97.50^{\pm6.22}$ |
| Att-Feat-C | $49.00^{\pm3.26}$ | $50.89^{\pm3.26}$ | $57.67^{\pm3.22}$ | $54.44^{\pm3.25}$ | $77.83^{\pm3.32}$ | $66.00^{\pm3.09}$ | $82.67^{\pm2.47}$ | $100.00^{\pm4.38}$ |
| Att-Spa-R | $59.56^{\pm4.52}$ | $52.44^{\pm4.59}$ | $64.00^{\pm4.42}$ | $49.33^{\pm4.60}$ | $78.00^{\pm4.67}$ | $77.33^{\pm3.86}$ | $78.44^{\pm3.79}$ | $97.50^{\pm6.22}$ |
| Att-Spa-C | $52.00^{\pm3.26}$ | $61.78^{\pm3.17}$ | $59.33^{\pm3.20}$ | $61.56^{\pm3.17}$ | $81.83^{\pm3.08}$ | $64.56^{\pm3.12}$ | $74.67^{\pm2.84}$ | $100.00^{\pm4.38}$ |
| Mem-Dis-Cat-R | $86.00^{\pm2.27}$ | $71.89^{\pm2.93}$ | $44.00^{\pm3.24}$ | $78.56^{\pm2.68}$ | $85.50^{\pm2.82}$ | $87.67^{\pm2.15}$ | $88.33^{\pm2.10}$ | $92.50^{\pm8.64}$ |
| Mem-Dis-Cat-C | $53.11^{\pm2.66}$ | $75.26^{\pm2.30}$ | $69.78^{\pm2.45}$ | $77.78^{\pm2.22}$ | $89.80^{\pm1.98}$ | $85.19^{\pm1.89}$ | $91.04^{\pm1.53}$ | $97.50^{\pm6.22}$ |
| Mem-Dis-Loc-R | $56.00^{\pm3.24}$ | $30.89^{\pm3.01}$ | $34.33^{\pm3.10}$ | $34.11^{\pm3.09}$ | $63.61^{\pm3.84}$ | $63.11^{\pm3.15}$ | $94.33^{\pm1.52}$ | $100.00^{\pm4.38}$ |
| Mem-Dis-Loc-C | $52.00^{\pm2.66}$ | $50.67^{\pm2.66}$ | $55.33^{\pm2.65}$ | $51.41^{\pm2.66}$ | $54.18^{\pm3.25}$ | $51.33^{\pm2.66}$ | $77.63^{\pm2.22}$ | $90.00^{\pm9.55}$ |

Table A.4.2: Average granular accuracy (%) comparison of Qwen2.5-VL-7B using different captioning methods. Scores are presented as mean percentage accuracy ($\pm$ standard deviation).

| Task | Base | SC | SC-I | PC | One Shot |
|---|---|---|---|---|---|
| Mem-Cat-R | $76.53^{\pm3.03}$ | $80.00^{\pm2.86}$ | $69.60^{\pm4.02}$ | $85.87^{\pm2.49}$ | $78.53^{\pm2.93}$ |
| Mem-Cat-C | $82.13^{\pm2.74}$ | $91.07^{\pm2.05}$ | $76.40^{\pm3.71}$ | $93.87^{\pm1.73}$ | $86.80^{\pm2.42}$ |
| Mem-Loc-R | $33.20^{\pm3.36}$ | $72.00^{\pm3.21}$ | $63.40^{\pm4.21}$ | $97.47^{\pm1.15}$ | $32.27^{\pm3.34}$ |
| Mem-Loc-C | $52.13^{\pm3.57}$ | $69.73^{\pm3.28}$ | $59.20^{\pm4.29}$ | $96.00^{\pm1.42}$ | $50.80^{\pm3.57}$ |
| Perc-Cat-R | $74.67^{\pm6.90}$ | $76.67^{\pm6.72}$ | $65.00^{\pm9.19}$ | $80.67^{\pm6.29}$ | $76.67^{\pm6.72}$ |
| Perc-Cat-C | $92.67^{\pm4.25}$ | $93.33^{\pm4.09}$ | $90.00^{\pm5.96}$ | $82.67^{\pm6.04}$ | $90.00^{\pm4.84}$ |
| Perc-Loc-R | $33.33^{\pm7.46}$ | $73.33^{\pm7.01}$ | $58.00^{\pm9.50}$ | $99.33^{\pm1.78}$ | $27.33^{\pm7.07}$ |
| Perc-Loc-C | $55.33^{\pm7.86}$ | $72.67^{\pm7.07}$ | $51.00^{\pm9.61}$ | $99.33^{\pm1.78}$ | $44.00^{\pm7.85}$ |
| CVR-Cat-H | $39.33^{\pm5.49}$ | $45.33^{\pm5.60}$ | $46.00^{\pm6.84}$ | $52.33^{\pm5.62}$ | $31.33^{\pm5.22}$ |
| CVR-Loc-H | $29.33^{\pm5.13}$ | $46.67^{\pm5.61}$ | $36.50^{\pm6.61}$ | $61.33^{\pm5.48}$ | $30.00^{\pm5.16}$ |
| CVR-Cat-M | $54.00^{\pm7.88}$ | $57.33^{\pm7.82}$ | $61.00^{\pm9.39}$ | $66.67^{\pm7.46}$ | $58.00^{\pm7.80}$ |
| CVR-Loc-M | $49.33^{\pm7.90}$ | $59.33^{\pm7.77}$ | $52.00^{\pm9.61}$ | $60.00^{\pm7.75}$ | $44.00^{\pm7.85}$ |
| CVR-Cat-L | $60.00^{\pm7.75}$ | $62.67^{\pm7.65}$ | $61.00^{\pm9.39}$ | $61.33^{\pm7.70}$ | $61.33^{\pm7.70}$ |
| CVR-Loc-L | $56.00^{\pm7.85}$ | $64.67^{\pm7.56}$ | $56.00^{\pm9.55}$ | $68.67^{\pm7.34}$ | $58.00^{\pm7.80}$ |
| Att-Feat-R | $58.67^{\pm4.53}$ | $80.44^{\pm3.66}$ | $46.00^{\pm5.60}$ | $99.78^{\pm0.60}$ | $58.89^{\pm4.53}$ |
| Att-Feat-C | $54.44^{\pm3.25}$ | $76.56^{\pm2.76}$ | $58.17^{\pm3.93}$ | $67.22^{\pm3.06}$ | $54.00^{\pm3.25}$ |
| Att-Spa-R | $49.33^{\pm4.60}$ | $72.22^{\pm4.13}$ | $40.67^{\pm5.52}$ | $83.78^{\pm3.40}$ | $51.11^{\pm4.60}$ |
| Att-Spa-C | $61.56^{\pm3.17}$ | $77.33^{\pm2.73}$ | $63.33^{\pm3.84}$ | $57.11^{\pm3.23}$ | $61.89^{\pm3.17}$ |
| Mem-Dis-Cat-R | $78.56^{\pm2.68}$ | $79.56^{\pm2.63}$ | $66.00^{\pm3.78}$ | $75.44^{\pm2.81}$ | $77.67^{\pm2.72}$ |
| Mem-Dis-Cat-C | $77.78^{\pm2.22}$ | $83.33^{\pm1.99}$ | $78.56^{\pm2.68}$ | $85.56^{\pm1.88}$ | $70.89^{\pm2.42}$ |
| Mem-Dis-Loc-R | $34.11^{\pm3.09}$ | $75.56^{\pm2.80}$ | $65.67^{\pm3.79}$ | $94.44^{\pm1.51}$ | $29.56^{\pm2.98}$ |
| Mem-Dis-Loc-C | $51.41^{\pm2.66}$ | $70.81^{\pm2.42}$ | $59.89^{\pm3.20}$ | $89.93^{\pm1.61}$ | $52.22^{\pm2.66}$ |

Table A.4.3: Cognitive benchmark tasks, their abbreviations, and chance levels.

| Task | Abbreviation | Chance Level (%) |
|---|---|---|
| Perception Category Report | Perc-Cat-R | 13 |
| Perception Category Compare | Perc-Cat-C | 50 |
| Perception Location Report | Perc-Loc-R | 25 |
| Perception Location Compare | Perc-Loc-C | 50 |
| Feature Attention Report | Att-Feat-R | 25 |
| Feature Attention Compare | Att-Feat-C | 50 |
| Spatial Attention Report | Att-Spa-R | 13 |
| Spatial Attention Compare | Att-Spa-C | 50 |
| Memory Category Report | Mem-Cat-R | 13 |
| Memory Category Compare | Mem-Cat-C | 50 |
| Memory Location Report | Mem-Loc-R | 25 |
| Memory Location Compare | Mem-Loc-C | 50 |
| Memory with Distractors Category Report | Mem-Dis-Cat-R | 13 |
| Memory with Distractors Category Compare | Mem-Dis-Cat-C | 50 |
| Memory with Distractors Location Report | Mem-Dis-Loc-R | 25 |
| Memory with Distractors Location Compare | Mem-Dis-Loc-C | 50 |
| Composite Visual Reasoning Category High | CVR-Cat-H | 7 |
| Composite Visual Reasoning Location High | CVR-Loc-H | 7 |
| Composite Visual Reasoning Category Medium | CVR-Cat-M | 50 |
| Composite Visual Reasoning Location Medium | CVR-Loc-M | 50 |
| Composite Visual Reasoning Category Low | CVR-Cat-L | 50 |
| Composite Visual Reasoning Location Low | CVR-Loc-L | 50 |

Table A.4.4: The iWISDM `AutoTask` parameters used to generate the PAM tasks. No and/or operators, boolean operators and switch operators are allowed in these tasks.

| Complexity | # of trial frames | # of distractors | root operators |
|---|---|---|---|
| Perception Category Report | 1 | 0 | GetCategory |
| Perception Category Compare | 2 | 0 | IsSame |
| Perception Location Report | 1 | 0 | GetLoc |
| Perception Location Compare | 2 | 0 | IsSame |
| Feature Attention Report | 1 | 1-3 | GetLoc |
| Feature Attention Compare | 2 | 1-7 | IsSame |
| Spatial Attention Report | 1 | 1-3 | GetCategory |
| Spatial Attention Compare | 2 | 1-7 | IsSame |
| Memory Category Report | 2-6 | 0 | GetCategory |
| Memory Category Compare | 3-7 | 0 | IsSame |
| Memory Location Report | 2-6 | 0 | GetLoc |
| Memory Location Compare | 3-7 | 0 | IsSame |
| Memory with Distractors Category Report | 2-7 | 1-6 | GetCategory |
| Memory with Distractors Category Compare | 3-11 | 1-9 | IsSame |
| Memory with Distractors Location Report | 2-7 | 1-6 | GetLoc |
| Memory with Distractors Location Compare | 3-11 | 1-9 | IsSame |

Table A.4.5: The iWISDM `AutoTask` parameters used to generate the CVR tasks. The Cat and Loc task variants are obtained by allowing object feature selection of either Category or Location.

| Complexity | # of allowed and/or operators in task | # of switch operators | # of trial frames | # of distractors | root operators | boolean operators |
|---|---|---|---|---|---|---|
| Low | 1 | 0 | 6 | 0 | IsSame, And, Or, NotSame | IsSame, And, Or, NotSame |
| Medium | 1 | 1 | 8 | 0 | IsSame, And, Or, NotSame | IsSame, And, Or, NotSame |
| High | 1-2 | 1 | 9 | 0 | IsSame, And, Or, NotSame, GetLoc, GetCategory | IsSame, And, Or, NotSame |
| High-Distractor | 1-2 | 1 | 12 | 4 | IsSame, And, Or, NotSame, GetLoc, GetCategory | IsSame, And, Or, NotSame |

## A.5 Captioning Method Validation

Table A.5.1: PAM and CVR performance of Qwen2.5-VL-72B using different captioning methods. The Base column shows absolute percentage accuracies ($\pm$ standard deviation). Subsequent columns show the percentage change from the Base method. Positive changes are shown in green, negative in red. SC: Self-captioning, PC: Pre-captioned. Asterisks denote statistical significance: * $p < 0.05$, ** $p < 0.01$, *** $p < 0.001$.

| Task | Qwen 72B Base | Qwen SC | Qwen PC |
|---|---|---|---|
| Perception (Cat) | $89.00^{\pm4.36}$ | $-2.67^{\pm3.89}$ | $+2.00^{\pm4.00}$ |
| Perception (Loc) | $70.50^{\pm6.27}$ | $+9.17^{\pm4.54}$ | $+29.50^{\pm0.94}$ |
| Feature Attention | $80.44^{\pm2.59}$ | $+8.15**^{\pm1.70}$ | $+19.22***^{\pm0.43}$ |
| Spatial Attention | $80.56^{\pm2.58}$ | $+5.44**^{\pm1.85}$ | $+15.44***^{\pm1.29}$ |
| Memory (Cat) | $87.69^{\pm1.29}$ | $-1.42*^{\pm1.10}$ | $+6.59***^{\pm0.91}$ |
| Memory (Loc) | $58.44^{\pm1.97}$ | $+17.78***^{\pm1.39}$ | $+36.47***^{\pm0.88}$ |
| CVR-Cat-L | $80.00^{\pm7.77}$ | $-4.67^{\pm6.84}$ | $+20.00^{\pm1.85}$ |
| CVR-Loc-L | $52.00^{\pm9.61}$ | $+30.67^{\pm6.04}$ | $+47.00^{\pm2.64}$ |
| CVR-Cat-M | $88.00^{\pm6.41}$ | $-11.33^{\pm6.72}$ | $+6.00^{\pm4.85}$ |
| CVR-Loc-M | $48.00^{\pm9.61}$ | $+28.67^{\pm6.72}$ | $+42.00^{\pm5.96}$ |
| CVR-Cat-H | $69.83^{\pm6.33}$ | $-7.83^{\pm5.46}$ | $+8.17^{\pm5.71}$ |
| CVR-Loc-H | $40.97^{\pm6.74}$ | $+24.36^{\pm5.35}$ | $+50.03^{\pm4.00}$ |

Table A.5.2: PAM and CVR performance of GPT-4o using different captioning methods. The Base column shows absolute percentage accuracies ($\pm$ standard deviation). Subsequent columns show the percentage change from the Base method. Positive changes are shown in green, negative in red. SC: Self-captioning, PC: Pre-captioned. Asterisks denote statistical significance: * p < 0.05, ** p < 0.01, *** p < 0.001.

| Task | 4o Base | 4o SC | 4o PC |
|---|---|---|---|
| Perception (Cat) | $90.33^{\pm3.36}$ | $-2.00^{\pm3.64}$ | $+0.67^{\pm3.26}$ |
| Perception (Loc) | $89.00^{\pm3.55}$ | $+8.33^{\pm1.91}$ | $+11.00*^{\pm0.63}$ |
| Feature Attention | $87.19^{\pm1.78}$ | $+8.89***^{\pm1.04}$ | $+12.81***^{\pm0.14}$ |
| Spatial Attention | $75.93^{\pm2.28}$ | $+15.33***^{\pm1.51}$ | $+19.85***^{\pm1.08}$ |
| Memory (Cat) | $91.47^{\pm0.89}$ | $+1.73^{\pm0.81}$ | $+3.44***^{\pm0.70}$ |
| Memory (Loc) | $83.47^{\pm1.21}$ | $+9.89***^{\pm0.81}$ | $+12.22***^{\pm0.66}$ |
| CVR-Cat-L | $91.33^{\pm4.56}$ | $+2.00^{\pm4.09}$ | $+8.67^{\pm1.25}$ |
| CVR-Loc-L | $66.00^{\pm7.50}$ | $+27.33^{\pm4.09}$ | $+34.00^{\pm1.25}$ |
| CVR-Cat-M | $96.67^{\pm3.07}$ | $-7.33^{\pm4.98}$ | $+0.67^{\pm2.81}$ |
| CVR-Loc-M | $82.67^{\pm6.04}$ | $+7.33^{\pm4.84}$ | $+14.67^{\pm2.81}$ |
| CVR-Cat-H | $83.67^{\pm4.18}$ | $-2.00^{\pm4.37}$ | $+2.67^{\pm3.89}$ |
| CVR-Loc-H | $64.67^{\pm5.38}$ | $+18.67*^{\pm4.21}$ | $+30.33*^{\pm2.52}$ |

## A.6 Additional Fine-tuning Results

Table A.6.1: PAM and CVR performance of Qwen2.5-VL-32B base model and fine-tuned variants using LoRA with different amounts of training data. The Base column shows absolute percentage accuracies ($\pm$ standard deviation). Subsequent columns show the percentage change in accuracy from the Base method ($\pm$ standard deviation). Positive changes are shown in green, and negative in red. Asterisks denote statistical significance: * p < 0.05, ** p < 0.01, *** p < 0.001.

| Task | 32B Base | 32B LoRA 1k | 32B LoRA 100k |
|---|---|---|---|
| Memory (Cat) | $85.38^{\pm1.13}$ | $-1.91^{\pm1.19}$ | $+2.61***^{\pm1.04}$ |
| Memory (Loc) | $48.89^{\pm1.63}$ | $+8.44***^{\pm1.61}$ | $+16.80***^{\pm1.55}$ |
| Perception (Cat) | $87.67^{\pm3.73}$ | $-1.33^{\pm3.89}$ | $-1.00^{\pm3.85}$ |
| Perception (Loc) | $49.67^{\pm5.62}$ | $+19.33^{\pm5.21}$ | $+29.67*^{\pm4.57}$ |
| Feature Attention | $76.13^{\pm2.27}$ | $-11.13*^{\pm2.54}$ | $-10.31*^{\pm2.53}$ |
| Spatial Attention | $80.98^{\pm2.09}$ | $-0.54^{\pm2.12}$ | $+1.52^{\pm2.03}$ |
| CVR-Cat-H | $59.39^{\pm5.52}$ | $-6.06^{\pm5.61}$ | $+16.94*^{\pm4.79}$ |
| CVR-Loc-H | $35.33^{\pm5.38}$ | $+5.33^{\pm5.52}$ | $+26.33*^{\pm5.47}$ |
| CVR-Cat-M | $72.67^{\pm7.07}$ | $-19.33^{\pm7.88}$ | $-4.67^{\pm7.38}$ |
| CVR-Loc-M | $46.00^{\pm7.88}$ | $+6.00^{\pm7.89}$ | $+15.33^{\pm7.70}$ |
| CVR-Cat-L | $83.33^{\pm5.95}$ | $-24.67^{\pm7.78}$ | $-18.00^{\pm7.53}$ |
| CVR-Loc-L | $48.67^{\pm7.90}$ | $+10.00^{\pm7.78}$ | $+15.33^{\pm7.59}$ |

## A.7 PAM & CVR vs MMMU-Pro Performance

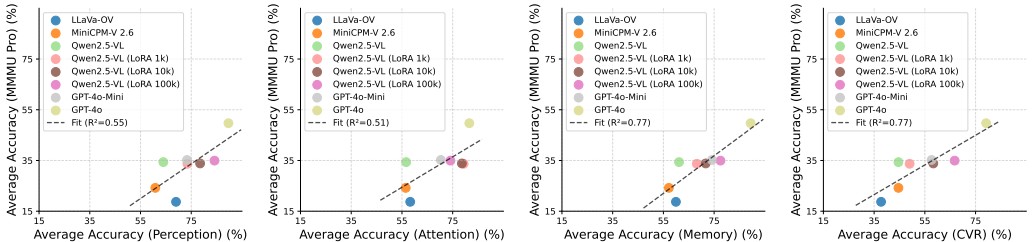

Figure A.7.1: Scatter plots comparing average PAM task (Perception, Attention, Memory) performance and CVR task performance against MMMU-Pro performance across all models. Each point represents a different model configuration. The x-axis shows the average accuracy on the specified PAM task category (averaging Loc and Cat), and the y-axis shows the overall MMMU-Pro accuracy.

## A.8 PAM & CVR vs VQAv2 Performance

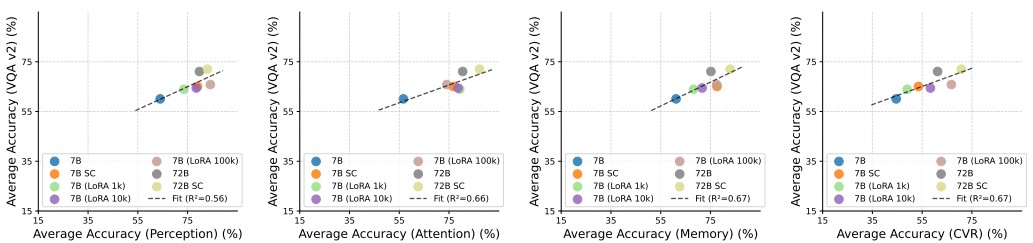

Figure A.8.1: Scatter plots comparing average PAM task (Perception, Attention, Memory) performance and CVR task performance against VQAv2 performance across different variants of Qwen2.5 VL models. Each point represents a different configuration. The x-axis shows the average accuracy on the specified PAM task category (averaging Loc and Cat), and the y-axis shows the overall VQAv2 accuracy.

## A.9 Attention Image Capacity Analysis

To investigate potential attention capacity limitations introduced by the presence of images during reasoning, we compared the average attention allocated to ground-truth caption tokens across the chain-of-thought (CoT) tokens generated by Qwen2.5-VL-7B in both PC (pre-captioned) and PC-I (pre-captioned with images) trials. Appendix Figure A.9.1 shows the inclusion of images significantly reduces the attention a given ground-truth caption token receives while the model outputs a CoT. These findings may reflect a potential attention capacity issue imposed by images during reasoning, which improvements during CoT training could remedy.

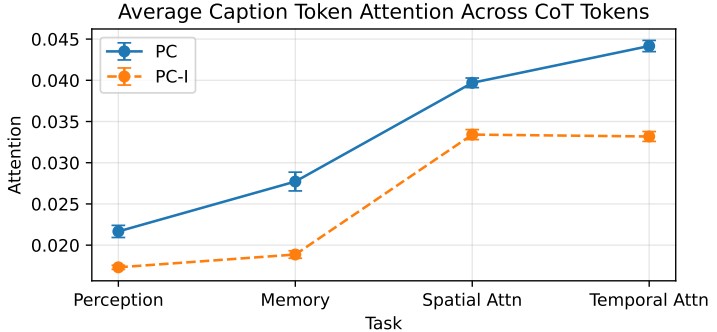

Figure A.9.1: Comparison of average Qwen2.5-VL-7b attention scores for ground-truth caption tokens across CoT tokens during PC and PC-I prompts of PAM tasks.

## A.10 Inference Runtime Analysis

To assess the computational overhead of our proposed Self-Captioning (SC) method, we measured the average inference runtime and compared it against the baseline direct-input method. We conducted these analyses on a Qwen2.5-VL-7B model running on a single NVIDIA A5000 GPU. We timed 10 trials for each task. The results, detailed in Table A.10.1, show that the SC method incurs a significant runtime cost, taking approximately twice as long as the base method for both simple and complex tasks.

Table A.10.1: Inference runtime comparison for the Self-Captioning (SC) vs. Base method on Qwen2.5-VL-7B. Runtimes are reported in seconds ($\pm$ standard deviation).

| Task | Base | SC |
|------|------|------|
| Mem-Cat-R | $1.81 \pm 1.46$ | $5.55 \pm 2.13$ |
| Mem-Cat-C | $2.06 \pm 1.86$ | $5.37 \pm 1.59$ |
| Mem-Loc-R | $1.60 \pm 1.20$ | $5.27 \pm 1.99$ |
| Mem-Loc-C | $2.20 \pm 1.72$ | $5.40 \pm 1.29$ |
| Perc-Cat-R | $1.54 \pm 1.16$ | $3.21 \pm 0.59$ |
| Perc-Cat-C | $1.96 \pm 1.43$ | $4.61 \pm 0.60$ |
| Perc-Loc-R | $1.54 \pm 0.90$ | $2.66 \pm 1.04$ |
| Perc-Loc-C | $2.07 \pm 1.13$ | $3.69 \pm 0.39$ |
| CVR-Cat-H | $6.58 \pm 4.67$ | $14.66 \pm 2.86$ |
| CVR-Loc-H | $6.33 \pm 5.12$ | $14.62 \pm 2.60$ |
| CVR-Cat-M | $4.76 \pm 4.05$ | $12.59 \pm 1.94$ |
| CVR-Loc-M | $6.70 \pm 4.12$ | $13.06 \pm 2.10$ |
| CVR-Cat-L | $3.58 \pm 3.29$ | $10.54 \pm 2.17$ |
| CVR-Loc-L | $2.83 \pm 3.12$ | $10.76 \pm 1.55$ |
| Att-Feat-R | $2.74 \pm 1.72$ | $4.06 \pm 0.89$ |
| Att-Feat-C | $2.91 \pm 1.59$ | $6.09 \pm 1.80$ |
| Att-Spa-R | $3.06 \pm 1.90$ | $4.66 \pm 1.27$ |
| Att-Spa-C | $2.78 \pm 1.52$ | $5.89 \pm 1.46$ |
| Mem-Dis-Cat-R | $3.31 \pm 2.49$ | $6.96 \pm 2.55$ |
| Mem-Dis-Cat-C | $3.21 \pm 3.18$ | $9.12 \pm 2.89$ |
| Mem-Dis-Loc-R | $2.77 \pm 2.45$ | $6.51 \pm 2.45$ |
| Mem-Dis-Loc-C | $3.28 \pm 2.92$ | $8.35 \pm 2.25$ |

## A.11 Prompt & Script Examples

```json
{
    "messages": [
        {
            "role": "user",
            "content": "In this task, we will show you a series of frame images. Each frame will
            either be blank (delay frame) or contain one or more 3D objects. The objects will
            always be from one of eight categories: benches, boats, cars, chairs, couches,
            lighting, planes, and tables. For each category, there are eight unique objects that
            could be used in the task. Any object sampled will be displayed as an image taken
            from a random viewing angle. The objects will be placed in one of four locations:
            top left, top right, bottom left, and bottom right. If there are multiple objects
            on a single frame, only one of them would be specified in the task instruction
            by either its location or its category. A written instruction will be provided.
            Your goal is to follow the instructions and answer the question contained in the
            instructions. Answers will always be one of the following: true, false, bottom right,
            bottom left, top left, top right, benches, boats, cars, chairs, couches, lighting,
            planes, tables.

            Please solve the following task:
            Task instruction: observe object 1 in frame 1, observe object 2 in frame 2,
            observe object 3 in frame 3, observe object 4 in frame 4, observe object 5
            in frame 5, delay, observe object 6 in frame 7, delay, observe object 7 in frame 9,
            if identity of object 3 equals identity of object 2, then location of object 7
            not equals location of object 6 and identity of object 5 equals identity of object 4?
            else location of object 1?

            Here are the corresponding frames: <image><image><image><image><image><image><image>
            <image><image>

            What is the correct answer to this task? (bottom right, bottom left, top left, top right).
            Provide your answer here: "
        }, {
            "role": "assistant",
            "content": "top right"
        }
    ],
    "images": [
        "../trial0/frames/epoch0.png",
        "../trial0/frames/epoch0.png",
        "../trial0/frames/epoch0.png",
        "../trial0/frames/epoch0.png",
        "../trial0/frames/epoch0.png",
        "../trial0/frames/epoch0.png",
        "../trial0/frames/epoch0.png",
        "../trial0/frames/epoch0.png",
        "../trial0/frames/epoch0.png"
    ]
}
```

Figure A.11.1: Supervised LoRA Fine-tuning Prompt Example

```
"Frame 1: A chairs located at the top left",
"Frame 2: A chairs located at the top right",
"Frame 3: A benches located at the top left",
"Frame 4: A benches located at the bottom right",
"Frame 5: A boats located at the bottom left",
"Frame 6: A benches located at the bottom right",
"Frame 7: delay frame",
"Frame 8: delay frame",
"Frame 9: A planes located at the top left"
```

Figure A.11.2: Pre-caption example

```python
def caption_image(image_path, model, config):
    # Prepare a prompt asking the model to caption the image:
    caption_prompt = [
        {
            "role": "user",
            "content":[
                {
                    "type": "text",
                    "text": "Please provide a concise caption for the given image, \
                    including what the location of each the object in the images \
                    are and what the category of each object is. Each image either \
                    is blank (a delay frame) or contains one or more 3D objects from
                    ↪   \
                    one of eight categories: benches, boats, cars, chairs, couches,
                    ↪   \
                    lighting, planes, and tables. The object is placed in one of
                    ↪   four \
                    locations: top left, top right, bottom left, or bottom right."
                }, {
                "type": "image_url",
                "image_url": {"url":
                ↪   f"data:image/png;base64,{encode_image(image_path)}"}
                }
            ]

        },
    ]

    # Call the model to get a caption
    response = model.chat.completions.create(
        model=config.get("oai-model", "gpt-4o-mini"),
        messages=caption_prompt,
        max_tokens=1024
    )

    # The model's caption is expected in the response.
    caption = response.choices[0].message.content.strip()
    return caption
```

Figure A.11.3: Code for Self-captioning

```python
def evaluate_model(task_base, trial_num, model, config):
    trial_base = f"{task_base}/trial{trial_num}/frames/"
    task_info_path = os.path.join(trial_base, "new_task_info.json")

    with open(task_info_path, 'r') as file:
        data = json.load(file)

    # Extract the instruction and answers
    instruction = data['new_instruction']
    answers = data['answers']

    # Find the images
    pattern = 'epoch*.png'
    matching_files = [f for f in os.listdir(trial_base) if fnmatch.fnmatch(f,
    ↪    pattern)]
    matching_files.sort()

    images = [os.path.join(trial_base, file) for file in matching_files]

    # ----------------------------------------------------
    # NEW PART: First, caption each image
    # ----------------------------------------------------
    captions = []
    for image_path in images:
        caption = caption_image(image_path, model, config)
        captions.append(caption)

    # Now we have a list of captions, one per frame
    # Incorporate these captions into the final prompt
    # Instead of sending images to the model in the final task, we send their
    ↪    captions:

    # Prepare the final evaluation prompts
    # The prompt should now describe that we have textual descriptions of each
    ↪    frame:
    possible_answers = reduce(lambda x, y: f"{x}, {y}",
    ↪    get_all_possible_ans(answers[-1]))

    prompts = [
        f"""In this task, we will show you a series of frames described by captions
        ↪    instead of direct images. Each frame either is blank (a delay frame) or
        ↪    contains a 3D object from one of eight categories: benches, boats, cars,
        ↪    chairs, couches, lighting, planes, and tables. The object is placed in
        ↪    one of four locations: top left, top right, bottom left, or bottom
        ↪    right.

        A written instruction is provided, and you must follow that instruction to
        ↪    find the correct answer. The valid answers are always one of the
        ↪    following: true, false, bottom right, bottom left, top left, top right,
        ↪    benches, boats, cars, chairs, couches, lighting, planes, tables.

        Task instruction: {instruction}

        Here are the frame captions:
        """,
        "\n".join([f"Frame {i+1}: {cap}" for i, cap in enumerate(captions)]) +
        ↪    "\n\n" + \
        f" What is the correct answer to this task? ({possible_answers}). Think
        ↪    step-by-step, analyze each frame and provide your answer
        ↪    here:\nAnswers:\nLet's think step by step."
    ]
    ...
```

Figure A.11.4: Example for Evaluation (Part 1/2).

```
...
messages = [
    {
        "role": "user",
        "content": [
            {
                "type": "text",
                "text": prompts[0],
            },
            {
                "type": "text",
                "text": prompts[1],
            },
        ]
    }
]

print(messages)
instruction = instruction.join([f"Frame {i+1}: {cap}" for i, cap in enumerate(captions)])

# Get the response for the final evaluation
response = model.chat.completions.create(
    model=config.get("oai-model", "gpt-4o-mini"),
    messages=messages,
    max_tokens=config.get("max_new_tokens", 50),
)

return instruction, answers[-1], response.choices[0].message.content
```

Figure A.11.5: Example for Evaluation (Part 2/2).

