# OpenReview forum: "Caption This, Reason That: VLMs Caught in the Middle"
_NeurIPS.cc/2025/Conference — NeurIPS 2025 spotlight_

### Official Review · Reviewer_n5pe · 2025-06-21

**Clarity:** 2
**Significance:** 2
**Originality:** 2
**Rating:** 4
**Confidence:** 4

**Summary:**

This paper explores the visual cognitive abilities of Vision-Language Models (VLMs), focusing on perception, attention, and memory. The authors follow a four-step approach. First, they build two datasets, PAM and CVT, using a toolkit called iWISDOM. Second, they test top-performing VLMs on these datasets to find their weaknesses. Third, they analyze the reasons behind these weaknesses. Finally, they fine-tune the models to improve their cognitive abilities. Overall, the paper builds custom datasets, uses them to test VLMs, and draws conclusions based on the results.

**Questions:**

1. The PAM and CVR datasets are built using iWISDOM, but the paper does not explain how they differ from the original iWISDOM dataset. It remains unclear why the existing dataset cannot be used directly. More explanation is needed to justify the creation of these new datasets.

2. The perception ability of VLMs has already been extensively studied. The PAM and CVR datasets do not seem to provide significant new insights in this area. As for attention and memory, the tasks mostly involve referring to objects across single or multiple frames, which are still closely tied to perception rather than true reasoning. Therefore, the connection to reasoning ability is weak.

3. There are already several works that explore the reasoning capabilities of VLMs, such as [1] and [2]. The paper should discuss these related works, clarify how its approach differs, and explain why it is necessary to create new datasets or tasks focused on reasoning.

4. In Table 1, the performance of GPT-4o and human participants on CVR-Cat is confusing. Despite being labeled as a low-difficulty task, the results show performance worse than the medium-difficulty tasks. The paper lacks a detailed analysis and fails to provide convincing explanations for this inconsistency.

[1] Visual Abductive Reasoning

[2] The Abduction of Sherlock Holmes: A Dataset for Visual Abductive Reasoning

**Ethical Concerns:**

["NO or VERY MINOR ethics concerns only"]

**Final Justification:**

The author's response tackles my concern, and agrees with the other reviewers that this paper is more complete with newly added contents in rebuttal. I would increase my rating by +1.

**Limitations:**

The paper aims to study the cognitive abilities of VLMs, but the problem definition remains more focused on perception rather than reasoning. It does not specifically address key reasoning challenges in existing VLMs, such as counting. In addition, the PAM and CVT datasets are not described in sufficient detail, and the paper lacks essential information such as dataset statistics, creation process, and sample examples. The differences between these datasets and the original iWISDOM dataset should also be clearly explained.

**Paper Formatting Concerns:**

Figure 1 and figure 2 are too small, and not clear to see.

**Quality:**

2

**Strengths And Weaknesses:**

Strengths:

1. The paper attempts to study the cognitive abilities of Vision-Language Models (VLMs), which is an exploratory and meaningful direction.

Weaknesses:

1. The motivation is not clearly presented. In the introduction, after the background section (Lines 25–41), the authors claim that "VLMs struggle with multi-object reasoning tasks such as counting or identifying objects...". However, the rest of the paper does not clearly explain how their method addresses the counting problem.

2. The paper lacks a clear and detailed description of the PAM and CVR datasets in Section 3.2. There is no information available about the dataset statistics, the creation process of the datasets, or example cases. It’s also unclear why two separate datasets are needed.

3. The description of the CVR dataset (Lines 139–145) is too brief.

4. The paper claims it can "identify specific weaknesses" of VLMs (Line 60), but this is not clearly described.

5. The claim that the paper reveals the reasoning bottleneck of VLMs (Line 62) is also not clearly described.

6. Fine-tuning on a small dataset does not necessarily address cognitive reasoning issues if the training and testing follow the same distribution. To prove that fine-tuning improves general reasoning rather than just helping the model memorize the data, the train/test should be out-of-distribution data.

---

> ### Author Rebuttal · Authors · 2025-07-31
>
> We would like to first thank the reviewer for their thoughtful feedback and suggestions. Our responses to your comments are as follows:
>
> **The motivation is not clearly presented. In the introduction, after the background section (Lines 25–41), the authors claim that "VLMs struggle with multi-object reasoning tasks such as counting or identifying objects...". However, the rest of the paper does not clearly explain how their method addresses the counting problem.**
>
> To clarify, our aim for that section of the introduction was to introduce weaknesses in VLM reasoning that other studies have found. Reasoning is a broad term that encompasses many cognitive abilities, including counting objects. We did not intend to imply that counting would be directly addressed in our study, as this work focused on evaluating core cognitive abilities (PAM) as well as high-level decision making where those core abilities are used (CVR). We will be sure to update the introduction section to make this clearer to the reader.
>
> **The paper lacks a clear and detailed description of the PAM and CVR datasets in Section 3.2. There is no information available about the dataset statistics, the creation process of the datasets, or example cases. It’s also unclear why two separate datasets are needed.**
>
> Thank you for pointing this out. We agree that Section 3.2 would benefit from a more detailed presentation of both the PAM and CVR datasets. In the revised manuscript, we will add dataset statistics and clarify the creation process. We had already included examples for all PAM and CVR task types in Figures A.1.1-5. We also address the differences between PAM and CVR and why they are both necessary below in response to the reviewer's question of *The PAM and CVR datasets are built using iWISDOM…*.
>
> **The description of the CVR dataset (Lines 139–145) is too brief.**
>
> In the revised manuscript we will provide further details on the description of the CVR dataset, similar to how we describe it below in response to the reviewer's question of "*The PAM and CVR datasets are built using iWISDM…*". However, as CVR is a replication of the data used in the benchmark section of the iWISDM paper, we feel it is sufficient to have a brief explanation and list of details, and point to the iWISDM paper, if the readers are interested in further details.
>
> **The paper claims it can "identify specific weaknesses" of VLMs (Line 60), but this is not clearly described.**
>
> We already mention the two specific weaknesses revealed by our analyses (spatial localization and selective attention) in the list of contributions at the end of the Introduction (lines 60-61). Furthermore, sections 4.1 and 4.2 of our manuscript provide detailed descriptions of the VLM weaknesses exposed by the PAM and CVR datasets.
>
> **The claim that the paper reveals the reasoning bottleneck of VLMs (Line 62) is also not clearly described.**
>
> We appreciate the reviewer’s feedback, and we will update the manuscript to include a clearer description of the reasoning bottleneck we find. We will amend the end of Section 4.3 (lines 275-292), where we have our current descriptions, to more directly describe that the reasoning bottleneck is caused by an inability of SOTA VLMs to effectively use CoT to first caption images and then subsequently reason with the extracted information.
>
> **Fine-tuning on a small dataset does not necessarily address cognitive reasoning issues if the training and testing follow the same distribution. To prove that fine-tuning improves general reasoning rather than just helping the model memorize the data, the train/test should be out-of-distribution data.**
>
> We agree entirely, and this is precisely why we made sure to evaluate the finetuned models on other cognitive benchmarks like MMMU-Pro and MMBench. You can see in the Appendix Table A.3.2 that slight improvements to these benchmarks were made. Moreover, following suggestions from reviewers *o9hF*, *Dy9Q*, and *ptot*, we further tested the models on the VQAv2 dataset consisting of diverse questions about natural images, completely different from those used during fine-tuning. A gist of these results can be seen in our response to reviewer *Dy9Q*’s comment on "*Generalisation. Has SC/LoRA been tried on natural photographs (e.g., VQAv2) to confirm transfer beyond synthetic scenes?*". Altogether, these results confirm that fine-tuning improves the model’s performance on multiple out-of-distribution datasets. We will be sure to make these results more evident in the main text.
>
> **Questions:**
>
> **The PAM and CVR datasets are built using iWISDOM, but the paper does not explain how they differ from the original iWISDOM dataset. It remains unclear why the existing dataset cannot be used directly. More explanation is needed to justify the creation of these new datasets.**
>
> To clarify, iWISDM is an environment where datasets for visual decision-making tasks can be generated randomly or constructed manually, rather than a set of fixed datasets. To perform the experiments and analyses that our paper focuses on, we created PAM and CVR using iWISDM:
>
> - PAM is manually designed, explicitly targeting core cognitive abilities: Perception, Attention, and Memory. The Attention dataset also included the use of distractor objects, a feature of iWISDM that was not explicitly used in the iWISDM paper.
>
>
> - CVR employs the iWISDM *AutoTask* environment to dynamically generate complex tasks via random compositions of basic task operators. These tasks are specifically generated to *assess the general visual reasoning capacity of different models* in our work. The tasks in CVR were randomly generated following the same parameters as used in the iWISDM paper for its section on benchmarking.
>
> While necessary in achieving the main contributions of this study, PAM and CVR were not the main contributions in themselves. We will clarify this and provide further details on the dataset distinctions in the revised manuscript.
>
> **The perception ability of VLMs has already been extensively studied. The PAM and CVR datasets do not seem to provide significant new insights in this area. As for attention and memory, the tasks mostly involve referring to objects across single or multiple frames, which are still closely tied to perception rather than true reasoning. Therefore, the connection to reasoning ability is weak.**
>
> We acknowledge that the perceptual abilities of VLMs have been studied previously. However, our work offers a new perspective in several key ways: 1) it reveals substantial failures in object localization, even in simple scenes with few objects and no background; 2) it demonstrates that a simple self-captioning strategy can meaningfully address this limitation across multiple visual reasoning tasks; and 3) it grounds evaluation in a cognitively inspired framework, using tasks from classic studies of perception, memory, and attention. Together, we believe these contributions offer a fresh lens on the cognitive capabilities of VLMs that, to our knowledge, has not been explored in prior work.
>
> **There are already several works that explore the reasoning capabilities of VLMs, such as [1] and [2]. The paper should discuss these related works, clarify how its approach differs, and explain why it is necessary to create new datasets or tasks focused on reasoning.**
>
> We thank the reviewer for pointing us to these papers and will include them in our discussion of reasoning in VLMs in the revised manuscript. To clarify the differences: both papers focus on visual *abductive* reasoning, which differs from our work in key ways. (1) Abductive reasoning involves inferring the most plausible explanation—often a hidden cause—for a given observation. In contrast, our work focuses on *logical* reasoning, where VLMs are required to follow explicit instructions with unambiguous answers, leaving no room for alternative interpretations. (2) Unlike those works, our tasks involve *multiple images* and require *explicit relational reasoning* across them. Existing multi-image datasets are limited in number and typically involve only two images per task, whereas our tasks operate over larger sets.
>
> **In Table 1, the performance of GPT-4o and human participants on CVR-Cat is confusing. Despite being labeled as a low-difficulty task, the results show performance worse than the medium-difficulty tasks. The paper lacks a detailed analysis and fails to provide convincing explanations for this inconsistency.**
>
> We acknowledge the reviewer's observation regarding the unexpected performance results in Table 1. We had briefly noted this anomaly in the original manuscript (line 239) with a possible explanation related to GPT-4o. On further reflection, we believe this counterintuitive result for human participants may also stem from the presence of the if-then-else operator in Medium complexity CVR-Cat tasks. Although Medium tasks involve more logical operations overall, the structured nature of if-then-else may restrict the number of objects and comparisons that need to be considered at a time, helping participants effectively segment instructions into manageable substeps.
>
> Relatedly, we also found that despite the apparent trend in accuracy across the three levels, the response times for human subjects consistently ramped up with higher complexity (averages: L=32.6s, M=39.1s, H=59.3s). These response times provide a more nuanced outlook on task difficulty. Despite having a higher response accuracy on medium CVR-Cat tasks than on the equivalent low-complexity tasks, subjects required more time to solve them. We will ensure that all response time data is included in the updated manuscript, along with a more detailed discussion of the specific result. We are also open to any explanations or approaches that the reviewer may have for further investigation.
>
> **Formatting Concerns:**
> Thank you for pointing out these issues. They have been corrected in the updated manuscript!

---

### Official Review · Reviewer_ptot · 2025-06-30

**Clarity:** 3
**Significance:** 2
**Originality:** 3
**Rating:** 4
**Confidence:** 3

**Summary:**

This paper systematically investigates Vision-Language Models (VLMs) through the lens of cognitive science, breaking down their abilities into Perception, Attention, and Memory (PAM) and assessing how these core capabilities impact composite reasoning. It introduces:

1. PAM and Composite Visual Reasoning (CVR) datasets, procedurally generated with iWISDM, to test fine-grained cognitive abilities.

2. A vision-text decoupling analysis, using self-captioning and pre-captioning to understand if failures come from perception or integration.

3. Experiments showing targeted fine-tuning significantly improves performance, especially on spatial reasoning.

**Questions:**

As mentioned in the weakness.

**Ethical Concerns:**

["NO or VERY MINOR ethics concerns only"]

**Final Justification:**

The response has addressed most of my previous concerns and questions. I increase my score by to 4.

**Limitations:**

Yes

**Quality:**

2

**Strengths And Weaknesses:**

### **Strengths**

1. **Novel Analytical Framework:**
   The paper reframes VLM evaluation around core cognitive primitives—**Perception, Attention, and Memory**—rather than simply task types. This perspective, inspired by cognitive science, provides clear, testable hypotheses about where and why models fail.

2. **Interesting Experimental Observations:**
   The work conducts several insightful experiments to probe specific deficiencies of current VLMs, such as the vision-text decoupling analyses that help disentangle perceptual encoding from reasoning limitations.

---

### **Weaknesses**

1. **Limited Evaluation Data Diversity:**
   All tasks are generated from a synthetic dataset with only 8 object categories and uniform backgrounds. This significantly limits ecological validity, as real-world images involve clutter, occlusion, and more diverse visual content.

2. **Domain-Specific Testing Setup:**
   The reliance on the iWISDM environment creates a highly specialized evaluation scenario. This makes it difficult to disentangle whether observed errors reflect true perception and reasoning deficits, or simply failures of out-of-distribution generalization.

3. **Interpretation of Fine-tuning Gains:**
   Regarding the claimed contribution (line 65), it is not surprising that fine-tuning on the target domain improves performance on tasks from that same domain. Therefore, it is hard to attribute these gains specifically to improved *diverse cognitive reasoning abilities*, rather than to domain memorization or overfitting.

---

> ### Author Rebuttal · Authors · 2025-07-31
>
> We would first like to thank the reviewer for their thoughtful feedback and suggestions! Our responses to your comments are as follows:
>
> **Limited Evaluation Data Diversity and Domain-Specific Testing Setup:**
>
> We chose iWISDM and synthetic objects because of their strong controllability, which allowed us to precisely control the object properties in the context of diverse cognitive tasks. With regard to the reviewer’s concern about ecological validity, we had already shown that the performance of different models on our PAM and CVR task sets is strongly correlated with those on benchmarks with natural images (MMMU-Pro). This suggests that the low performance of these models on our tasks is not incidental nor a result of poor data diversity.
>
> Regardless, to directly address the reviewer’s concern, we ran new experiments in which we tested our methods on the VQAv2 dataset. The results are as follows:
>
> | Model/Configuration   | VQAv2 (5k subset) Accuracy (%) |
> |------------------------|--------------|
> | 7B Base                  | 60.07        |
> | 7B SC                    | 65.07        |
> | 7B LoRA 1k            | 63.96        |
> | 7B LoRA 10k           | 64.44        |
> | 7B LoRA 100k          | 65.84        |
> |72B Base| 71.09 |
> |72B SC| 72.02  |
>
> As evident, an almost 5-6% improvement on VQAv2 from both our simple SC method and LoRA finetuning for the 7B models, and 1% improvement for the larger 72B model, which suggests that both of these approaches lead to general improvements in the VLMs’ ability to perform general visual reasoning. We will include these results along with other benchmarks in Table A3.2.
>
> **Interpretation of Fine-tuning Gains:**
>
> We included this claim based on two key results in the paper: 1) In Table 3, we showed that the fine-tuned QWEN2.5-VL-7b model performed substantially better on unseen tasks from PAM; 2) In Table 3 (reported on lines 312-315), we show that fine-tuning the QWEN2.5-VL-7b model led to slight improvements on MMBench and MMUPro benchmarks, both of which contain realistic images from different domains. In addition, the newly performed VQAv2 experiments and results show significant improvements from fine-tuning. Together, these results indicate that the gain in performance is large in tasks that are conceptually close to the training distribution, but somewhat remains in natural domains with relatively little overlap with the training distribution. To address the reviewer’s concern we will revise the claims (and related statements) to the following: “Demonstrating that targeted fine-tuning on diverse cognitive reasoning tasks can moderately enhance core cognitive abilities in VLMs *when tested with paradigms similar to those used to test humans*, while also generally validating that improvements on PAM and CVR tasks translate to broader gains in visual reasoning performance.”
>
> Finally, we will revise the abstract and the text and tone down our claim about the effect of finetuning by changing the claim from “substantially improving core cognitive abilities” to  “moderately improving core cognitive abilities”.

---

### Official Review · Reviewer_Dy9Q · 2025-07-03

**Clarity:** 3
**Significance:** 3
**Originality:** 3
**Rating:** 5
**Confidence:** 4

**Summary:**

The paper introduces PAM (Perception-Attention-Memory) and CVR (Composite Visual Reasoning)––two procedurally-generated suites that probe VLMs along cognitive-science axes. Seven state-of-the-art models (GPT-4o family, Qwen2.5-VL, MiniCPM-V, InternVL-2.5, LLaVA-OneVision) plus humans are evaluated. Results reveal strong object-category perception but large gaps in spatial reasoning and selective attention. A simple vision-text decoupling trick (self-captioning) substantially narrows the gap, and LoRA fine-tuning on 1k–100k CVR samples further boosts performance. PAM/CVR scores correlate tightly with MMMU-Pro, highlighting their diagnostic value.

**Questions:**

Questions for the Authors

- Human study details. How many annotators? Were images shuffled to reduce practice effects?
- Statistical significance. Are accuracy gaps tested (e.g., bootstrap CIs) or visually inspected only?
- Caption quality. For self-captioning, do hallucinated captions ever hurt reasoning, and how is that measured?
- Generalisation. Has SC/LoRA been tried on natural photographs (e.g., VQAv2) to confirm transfer beyond synthetic scenes?

**Ethical Concerns:**

["NO or VERY MINOR ethics concerns only"]

**Final Justification:**

The authors clarifications have addressed my questions well and have helped to further refine the paper. After careful consideration, I am pleased to increase my score by +1.

**Limitations:**

Limitations (some acknowledged, some additional)

- Synthetic bias. Only eight ShapeNet categories on plain backgrounds limit ecological validity.
- Single-frame captioning. SC assumes each frame can be described independently; coherence across complex videos remains untested.
- Overfitting risk. LoRA boosts CVR but slightly hurts MMMU-Pro at small data sizes; larger sets might magnify this trade-off.
- Attention-capacity hypothesis unproven. The paper infers attention interference from SC-I drop, but no probing of cross-modal token limits is shown.
- Missing energy/runtime analysis. Procedural tasks are cheap, yet large-scale SC + LoRA incurs extra inference cost.

**Paper Formatting Concerns:**

- Figure 1 caption split across columns; sub-figure labels (a/b) not referenced in text.
- Mixed citation styles—inline “[8–14]” vs. verbal forms—should follow NeurIPS guidelines.
- Line 155, Qwen2.5-VL-72b the b is in small case.

**Quality:**

3

**Strengths And Weaknesses:**

Paper Strengths

- Cognitive framing. Mapping VLM skills to classic P-A-M faculties is fresh and helps pinpoint bottlenecks.
- Procedurally generated benchmarks. Tasks of unlimited size avoid test-set saturation and enable controlled ablations.
- Comprehensive evaluation. Mix of open-source and proprietary models + human baseline yields a clear landscape.
- Actionable insight. Self-captioning is a one-line prompt change yet delivers 18–55 pp gains on spatial tasks.
- Practical fine-tuning study. Shows that even 1 k synthetic examples meaningfully lift location, attention, and memory scores.
- Correlation to real-world benchmarks. Strong R²≈0.9 between PAM/CVR and MMMU-Pro argues external validity.

---

> ### Author Rebuttal · Authors · 2025-07-31
>
> We would first like to thank the reviewer for their thoughtful feedback and suggestions! Our responses to your comments are as follows:
>
> **Questions:**
>
> **Human study details. How many annotators? Were images shuffled to reduce practice effects?**
>
> The details of the human study were included in Appendix A.2. To answer your question briefly, the study included 8 subjects and trials were all randomly generated to include random selection of object images.
>
> **Statistical significance. Are accuracy gaps tested (e.g., bootstrap CIs) or visually inspected only?**
>
> The sample size in our studies was limited (N=3 per model per subtask), resulting in low statistical power for some tasks, especially CVR and Perception tasks, they have 3 or 6 samples in total per task. However, in response to the reviewer’s concern, we performed significance tests for all of our main results.
>
> Briefly, we found that for Qwen 2.5 VL 7B, SC, SC-I PC contrasts with Base were 5/6 tasks, 2/6 tasks, 4/6 significant with at least p<0.05 on PAM, both SC and PC has a significant different (p < 0.05) on CVR-Loc-H with Base.
>
> For larger Qwen 2.5 VL 72B SC and PC contrast with Base in all Memory and Attention tasks with at least p < 0.05 significant.
>
> For closed model, GPT 4o, SC and PC contrast with base in 3/6 and 5/6 tasks, with p < 0.05 in PAM and both in CVR-Loc-H.
>
> Finally, the LoRA fine-tuned Qwen 2.5 VL 7B models, LoRA 1k, 10k, 100k demonstrated p < 0.05 differences in 4/6, 4/6, 6/6 tasks in PAM with the base model and all LoRA models have a p < 0.05 difference in CVR-CAR-H, 10k and 100k in CVR-Loc-H too.
>
> We will include the results in all tables in the revised manuscript.
>
> **Caption quality. For self-captioning, do hallucinated captions ever hurt reasoning, and how is that measured?**
>
> The answer is yes, and some results in the manuscript already address this. Specifically, the differences between PC and SC values reported in Tables 2 and A.5 capture the accuracy of VLMs when using ground-truth image captions (PC) versus VLM-generated captions (SC). We interpret the drop in performance between these conditions as an effect of captioning hallucinations. This is an important point that was not clearly stated, and we will clarify it in the revised manuscript.
>
> **Generalisation. Has SC/LoRA been tried on natural photographs (e.g., VQAv2) to confirm transfer beyond synthetic scenes?**
>
> That’s an excellent suggestion! Following the reviewer’s advice, we performed new experiments where we additionally tested the fine-tuned Qwen2.5-VL-7B and Qwen2.5-VL-72B on the VQAv2 dataset. The results for these tests:
>
> | Model/Configuration   | VQAv2 (5k subset) Accuracy (%) |
> |------------------------|--------------|
> | 7B Base                  | 60.07        |
> | 7B SC                    | 65.07        |
> | 7B LoRA 1k            | 63.96        |
> | 7B LoRA 10k           | 64.44        |
> | 7B LoRA 100k          | 65.84        |
> |72B Base| 71.09 |
> |72B SC| 72.02  |
>
> As evident, an almost 5-6% improvement on VQAv2 from both our simple SC method and LoRA finetuning for the 7B models, and 1% improvement for the larger 72B model, which suggests that both of these approaches lead to general improvements in the VLMs’ ability to perform general visual reasoning. We will include these results along with other benchmarks in Table A3.2.
>
> **Limitations**
>
> **Synthetic bias. Only eight ShapeNet categories on plain backgrounds limit ecological validity.**
>
> We chose iWISDM and synthetic objects because of their strong controllability, which allowed us to precisely control the object properties in the context of diverse cognitive tasks. With regard to the reviewer’s concern about ecological validity, we had already shown that the performance of different models on our PAM and CVR task sets is strongly correlated with those on benchmarks with natural images (MMMU-Pro). This suggests that the low performance of these models on our tasks is not incidental nor a result of poor data diversity.
>
> Further, as shown previously in response to the reviewer’s question about generalization to natural photographs, we validated our methods on the naturalistic benchmark VQAv2, and found a 5-6% gain in performance from both SC and LoRA.
>
> **Single-frame captioning. SC assumes each frame can be described independently; coherence across complex videos remains untested.**
>
> We acknowledge that this is a limitation of our current tests, as our experiments only involve video and do not consider more complex media formats such as videos. We will add this point to the limitations section of our paper to address the reviewer’s comment.
>
> **Overfitting risk. LoRA boosts CVR but slightly hurts MMMU-Pro at small data sizes; larger sets might magnify this trade-off.**
>
> We agree that LoRA fine-tuning comes with the risk of overfitting. However, on our largest training dataset size of 100k, LoRA actually led to marginal improvements on both MMBench and MMMU-Pro. Even more substantial were our additional results on VQAv2, probed from the reviewer’s comments on generalization, which show a ~5.8% improvement. With that said, fine-tuning via other methods, such as reinforcement learning or full SFT, would be interesting avenues for future experiments.
>
> **Attention-capacity hypothesis unproven. The paper infers attention interference from SC-I drop, but no probing of cross-modal token limits is shown.**
>
> We agree with the reviewer that we did not offer any evidence to support this statement and only speculated that this is likely due to this hypothesis. Following reviewer **o9hF**’s suggestion we examined the attention scores to probe the role of attention capacity in our observations. To address the reviewer’s suggestion, we compared the total attention scores on the relevant caption tokens between SC and SC-I experiments, visualized individual samples and also calculated the average values across many samples. We hypothesize that if the textual attention scores in SC-I are significantly lower than those in SC, that would confirm our hypothesis that the inclusion of images in the context distracts the model from the more reliable textual information. Our results show that the caption text tokens received a total of 0.022 attention averaged over the model's output when the SC method is used, and 0.017 when SC-I is used. This difference in attention supports our attention capacity hypothesis.
>
> *Note:* Unfortunately, we cannot provide visuals of these attention scores due to Neurips guidelines; however, we will add the results of these and further analyses to the Appendix.
>
> **Missing energy/runtime analysis. Procedural tasks are cheap, yet large-scale SC + LoRA incurs extra inference cost.**
>
> We thank the reviewer for suggesting this critical analysis we missed. We have now performed runtime analyses on 10 trials of a 1-frame task (Perc-Loc-R) and 10 trials of a 9-frame task (CVR-Loc-H) given to Qwen2.5-VL-7b. We find that the SC method is indeed slower than the standard method of input on average for both trial lengths. SC performed the 1-frame trials in an average of 5.731 seconds and the 9-frame trials in an average of 35.259 seconds. The base method performed the 1-frame trials in an average of 2.841 seconds and the 9-frame trials in an average of 13.410 seconds. These analyses revealed an important limitation to our method; despite the significant performance gains of SC, it comes at the trade-off of roughly a 2x slower run-time. We will be sure to include this limitation and the details of our run-time analysis in the manuscript.
>
> **Formatting Concerns**
>
> Thank you for pointing out these issues. They have been corrected in the updated manuscript!

---

> > ### Comment · Reviewer_Dy9Q · 2025-08-04
> >
> > Thank you to the authors for the detailed response. Your clarifications have addressed my questions well and have helped to further refine the paper. After careful consideration, I am pleased to increase my score by +1.

---

> > > ### Author Response · Authors · 2025-08-04
> > >
> > > We sincerely thank the reviewer for their positive re-evaluation. We are grateful that our clarifications were helpful and that the discussion has helped refine the paper. We will be sure to incorporate these points into the final version.

---

### Official Review · Reviewer_o9hF · 2025-07-03

**Clarity:** 3
**Significance:** 4
**Originality:** 3
**Rating:** 5
**Confidence:** 3

**Summary:**

The paper analyses the VLM performances across three core cognitive axis - perception, attention and memory. They perform rigorous experiments across seven models spanning from open-source models to GPT-4o. The authors analyse several gaps pertains these models in tasks such as spatial understanding and selective attention. The authors state the need for improved chain-of-thought capabilities. This works presents a holistic understanding of the VLM cognitive strengths, weakness to explore the bottlenecks in simultaneous perception and reasoning.

**Questions:**

The authors might refer to the weakness and answer the question presented.

**Ethical Concerns:**

["NO or VERY MINOR ethics concerns only"]

**Final Justification:**

The authors did address most of the points raised and thus the rating is revised.

**Limitations:**

Yes the authors provided a separate section for the Limitations

**Paper Formatting Concerns:**

No, the paper does not have any formatting issues.

**Quality:**

3

**Strengths And Weaknesses:**

Strengths:
1. VLM evalaution on Perception, Attention and Memory based on human faculties presents a well cognitive grounded framework.
2. The authors have well explored the VLM bottlenecks showing a consistent gaps in the spatial localisation and selective attention in almost all the models such as perception-loc, memory-loc and spatial-attention.
3. The authors also replaced images with gournd truth captions  to improve Qwen2.5-VL by 55pp on perception-loc and +22pp on feature attention validating that lan can reason when the visual information is provided in a text form.


Weakness:
1. The generalization of the fine-tuning of LoRA is limited to Qwen2.5-VL-7B., why not other larger models?
2. One of the major questions is relying on a single chain of thought template, the authors night explore few-shot prompting etc.
3. The authors could have tested on unseen real-image dataset VQAv2, as there is marginal change on MMBench and MMMU-Pro for LoRA finetuned model.
4. The authors do mention of LoRA overfitting and add dropout but might add a detailed ablation such as different dropout rates to quantify better.
5. The paper could have presented attention heat maps for attention capacity issues.

---

> ### Author Rebuttal · Authors · 2025-07-31
>
> **The generalization of the fine-tuning of LoRA is limited to Qwen2.5-VL-7B., why not other larger models?**
>
> We performed new fine-tuning experiments on Qwen2.5-VL-32b to address this request. The table below shows how fine-tuning on 1k and 10k datasets affected PAM and CVR performance.
>
> | Task               | 32B Base        | 32B LoRA 1k    | 32B LoRA 10k  |
> |--------------------|------------------|----------------|----------------|
> | Memory (Cat)       | 85.38 ± 1.13     | 83.47 ± 1.19   | 87.99 ± 1.04   |
> | Memory (Loc)       | 48.89 ± 1.63     | 57.33 ± 1.61   | 65.69 ± 1.55   |
> | Perception (Cat)   | 87.67 ± 3.73     | 86.33 ± 3.89   | 86.67 ± 3.85   |
> | Perception (Loc)   | 49.67 ± 5.62     | 69.00 ± 5.21   | 79.33 ± 4.57   |
> | Feature Attention  | 76.13 ± 2.27     | 65.00 ± 2.54   | 65.81 ± 2.53   |
> | Spatial Attention  | 80.98 ± 2.09     | 80.44 ± 2.12   | 82.51 ± 2.03   |
> | CVR-Cat-H          | 59.39 ± 5.52     | 53.33 ± 5.61   | 76.33 ± 4.79   |
> | CVR-Loc-H          | 35.33 ± 5.38     | 40.67 ± 5.52   | 61.67 ± 5.47   |
> | CVR-Cat-M          | 72.67 ± 7.07     | 53.33 ± 7.88   | 68.00 ± 7.38   |
> | CVR-Loc-M          | 46.00 ± 7.88     | 52.00 ± 7.89   | 61.33 ± 7.70   |
> | CVR-Cat-L          | 83.33 ± 5.95     | 58.67 ± 7.78   | 65.33 ± 7.53   |
> | CVR-Loc-L          | 48.67 ± 7.90     | 58.67 ± 7.78   | 64.00 ± 7.59   |
>
> Apart from a few outliers like CVR-Cat-M/H and Feature Attention, fine-tuning improved the model’s performance. Compared to the 7b variant, fine-tuning 32b had a similar but lesser effect. Unfortunately, due to time and computational constraints, we were unable to fully test our 32b models. The results for their performance on held-out iWISDM tasks, MMMU-Pro, MMBench, as well as a 100k LoRA model, will be added to the manuscript.
>
>
> **One of the major questions is relying on a single chain of thought template, the authors might explore few-shot prompting etc.**
>
> We had not included few- shot prompting in our experiments because of two reasons: 1) The context window of VLMs, as well as computational resource limits, severely restricted the number of images that could be included in the prompt. For this reason, we were unable to include more images as examples, especially for the complex tasks which already included up to 9 images; 2) Prior work had reported limited improvements in VLM reasoning capacity in few-shot settings [1,2].
>
> Nevertheless, to address the reviewer’s request more directly, we performed new one-shot experiments with Qwen2.5-VL-7B.
>
> | Task              | Base       | SC          | One Shot    | PC          |
> | ----------------- | ---------- | ----------- | ----------- | ----------- |
> | Memory (Cat)      | 78.59      | +4.72       | -1.55       | +6.27       |
> | Memory (Loc)      | 42.22      | +26.89      | -1.14       | +47.39      |
> | Perception (Cat)  | 83.67      | +1.33       | +2.33       | -2.00       |
> | Perception (Loc)  | 44.33      | +28.67      | -3.33       | +55.00      |
> | Feature Attention | 55.85      | +22.00      | -3.63       | +22.22      |
> | Spatial Attention | 57.48      | +18.15      | +0.74       | +8.52       |
> | CVR-Cat-H         | 39.33      | +6.00       | -10.33      | +13.00      |
> | CVR-Loc-H         | 29.33      | +17.33      | -2.33       | +32.00      |
> | CVR-Cat-M         | 54.00      | +3.33       | +8.00       | +12.67      |
> | CVR-Loc-M         | 49.33      | +10.00      | +6.67       | +10.67      |
> | CVR-Cat-L         | 60.00      | +2.67       | -2.00       | +1.33       |
> | CVR-Loc-L         | 56.00      | +8.67       | +2.00       | +12.67      |
>
> Compared with SC, one-shot didn't provide a strong performance boost. We will include these results as an additional table in the appendix.
>
> [1] Zong, Y., Bohdal, O., & Hospedales, T. (2025). VL-ICL Bench: The Devil in the Details of Multimodal In-Context Learning. ICLR 2025
>
> [2] Shukor, M., Rame, A., Dancette, C., & Cord, M. (2024). Beyond Task Performance: Evaluating and Reducing the Flaws of Large Multimodal Models with In-Context Learning. ICLR 2025
>
>
> **The authors could have tested on unseen real-image dataset VQAv2, as there is marginal change on MMBench and MMMU-Pro for LoRA finetuned model.**
>
> That’s an excellent suggestion! Following the reviewer’s advice, we performed new experiments where we additionally tested the fine-tuned Qwen2.5-VL-7B and Qwen2.5-VL-72B on the VQAv2 dataset. The results for these tests:
>
> | Model/Configuration   | VQAv2 (5k subset) Accuracy (%) |
> |------------------------|--------------|
> | 7B Base                  | 60.07        |
> | 7B SC                    | 65.07        |
> | 7B LoRA 1k            | 63.96        |
> | 7B LoRA 10k           | 64.44        |
> | 7B LoRA 100k          | 65.84        |
> |72B Base| 71.09 |
> |72B SC| 72.02  |
>
> This indicates an almost 5-6% improvement on VQAv2 from both our simple SC method and LoRA finetuning for the 7B models, and 1% improvement for the larger 72B model. These results further confirm that the observed improvements are robust and generalize to broader settings involving natural images.
>
> We will include these results along other benchmarks in Table A3.2.
>
> **The authors do mention of LoRA overfitting and add dropout but might add a detailed ablation such as different dropout rates to quantify better.**
>
> To find the best fine-tuning procedure, we ran a number of experiments during which we changed several hyperparameters, including the LR, LR scheduler decay ratio, dropout ratio, and optimizer parameters. Due to the sheer number of possible combinations, the high computational cost, and our limited academic-level resources, we were unable to try all possible combinations and therefore adopted the best setting for our smallest-scale experiments. Regarding dropout, we had briefly tried 0.0 and 0.1 before settling on 0.2. Our training runs with lower dropout ratios (0.0 and 0.1) experienced overfitting, resulting in a rise in the validation loss during training. These performance values will be added to the appendix to address the reviewer’s concern. Finally, we would like to note that the efficiency of the fine-tuning experiments was not directly tied to any of our main contributions.
>
> **The paper could have presented attention heat maps for attention capacity issues.**
>
> We thank the reviewer for their thoughtful suggestion. We believe examining the attention scores would be an effective way to probe the role of attention capacity in our observations. To address the reviewer’s suggestion, we will compare the total attention scores on the relevant caption tokens between SC and SC-I experiments, visualize individual samples and also present the average values across many samples. We hypothesize that if the textual attention scores in SC-I are significantly lower than those in SC, that would confirm our hypothesis that the inclusion of images in the context distracts the model from the more reliable textual information. Our results show that the caption text tokens received a total of 0.022 attention averaged over the model's output when the SC method is used, and 0.017 when SC-I is used. This difference in attention provides strong evidence in support of our attention capacity hypothesis.
>
> **Note:** we cannot provide visuals of these attention scores due to Neurips guidelines; however, we will add the results of these and further analyses to the Appendix.

---

> > ### Comment · Reviewer_o9hF · 2025-08-07
> > **Response to authors.**
> >
> > First and foremost, I would like to thank the authors for their response. Although the first point is still unclear as my question was to try using other models than Qwen2.5-VL-7B, but the authors did address rest of the points I have raise. I will be upgrading my rating by a point.

---

### Note · Authors · 2025-08-12

We sincerely thank the reviewers for their articulate and constructive feedback, and for engaging with our rebuttals in a positive and thoughtful manner. Their insights enabled us to perform essential new analyses and address previously overlooked issues. Almost every discussion led to additional experimental results that further strengthened our core claims. All new analyses and suggested changes will be incorporated into the updated manuscript.

**Summary of contributions**:
- Drawing inspiration from cognitive science methods for assessing human core cognitive abilities, we introduce a framework to reveal the cognitive profiles of state-of-the-art vision-language models.
- These profiles identified specific cognitive weaknesses, such as using spatial information and performing selective attention.
- Our analyses indicate that visual reasoning limitations primarily arise from challenges in integrating visual information, rather than from perceptual encoding or language reasoning alone. This suggests that current chain-of-thought training methods for visual reasoning remain largely ineffective, even in the strongest models.
- Based on this insight, we propose a simple yet highly effective text-image decoupling approach that improves visual reasoning not only on our new cognitive datasets, but also on other prominent vision-language benchmarks.
- We explore the effectiveness of fine-tuning on diverse cognitive task sets on downstream visual reasoning abilities. We show that improvements on our PAM and CVR datasets translate to broader gains in generalized visual reasoning performance.

*Finally, we would like to thank the Area Chair for facilitating a fair and respectful review process. We eagerly await your final decision.*

---

### Decision · Program_Chairs · 2025-09-17

**Decision:**

Accept (spotlight)

**Comment:**

**Summary**
This paper introduces a novel framework for evaluating VLMs with respect to isolated core cognitive abilities, by creating two procedurally generated benchmarks, PAM (Perception, Attention, Memory) and CVR (Composite Visual Reasoning). Benchmarking on seven SOTA models consistently reveal that while VLMs excel at object categorization, they exhibit significant weaknesses in spatial localization and selective attention. Vision-text decoupling analysis identifies the primary bottleneck not as flawed perception or language reasoning in isolation, but rather the VLM models' challenge in integrating visual information. To address this, the author team proposes a simple "Self-Captioning" prompting method that yields substantial gains, and demonstrate that targeted fine-tuning on synthetic cognitive tasks leads to improved performance that generalizes to established real-image benchmarks such as VQAv2.

**Strengths**
* Introduce a framework to reveal the cognitive profiles of SOTA VLMs that helps identify specific cognitive weaknesses (such as using spatial information and performing selective attention), drawing inspiration from cognitive science methods for assessing human core cognitive abilities.
* Analyses indicate that visual reasoning limitations primarily arise from challenges in integrating visual information, rather than from perceptual encoding or language reasoning alone. This suggests that current chain-of-thought training methods for visual reasoning remain largely ineffective, even in the strongest models.
* Based on this insight, propose a simple yet highly effective text-image decoupling approach that improves visual reasoning not only on our new cognitive datasets, but also on other prominent vision-language benchmarks. Also explored the effectiveness of fine-tuning on diverse cognitive task sets on downstream visual reasoning abilities, showing that improvements on the PAM and CVR datasets translate to broader gains in generalized visual reasoning performance.

**Potential Weaknesses, Rebuttal Discussion, and Reason for Accept/Reject**
The following key points were raised during the rebuttal period: Reviewer o9hF questioned the generalization of fine-tuning beyond the 7B model, and suggested testing on real-image datasets such as VQAv2 and exploring few-shot prompting; Reviewer Dy9Q requested details on human studies, statistical significance, an analysis of caption hallucinations, and runtime costs; Reviewer ptot expressed concerns about the synthetic data limiting ecological validity and the interpretation of fine-tuning gains; Reviewer n5pe requested clearer motivation, more dataset details, and a discussion of how the work differs from existing VLM reasoning studies.

The authors addressed these points comprehensively. They ran new experiments fine-tuning a 32B model, showing positive results. Crucially, they tested on VQAv2, revealing a 5-6% accuracy improvement from their methods, which strongly alleviates concerns about ecological validity and proves generalization to real-world images. They provided requested statistical analyses, runtime costs (~2x slower for self-captioning), and attention score analyses to support their capacity hypothesis. They also clarified the dataset descriptions, toned down certain claims, and added discussions of related work on reasoning.

All four reviewers were satisfied with the responses and increased their score ratings. The most critical concerns on generalization to diverse real-world settings and statistical rigor have been met with new experimental evidence. This paper should be a clear accept, as its core contributions of novel VLM evaluation framework & solution to a key perception-reasoning integration bottleneck stands even more strongly supported now.